

# Channelised, distributed, and disconnected: Spatial structure and temporal evolution of the subglacial drainage under a valley glacier in the Yukon

Camilo Andrés Rada Giacaman[1,2] and Christian Schoof[1]

[1]Department of Earth, Ocean and Atmospheric Sciences, University of British Columbia. 2207 Main Mall, Vancouver, BC Canada.
[2]Centro de Investigación GAIA Antárticah, Universidad de Magallanes. Avenida Bulnes 01855, Punta Arenas, Chile.

**Correspondence:** Camilo Rada (camilo@rada.cl)

**Abstract.** The subglacial drainage system is one of the main controls on basal sliding, but remains only partially understood. Here we expand the analysis of the eight-year dataset of borehole observations presented in Rada and Schoof (2018). These observations were made on a small, alpine polythermal valley glacier in the Yukon Territory, and were previously used to describe the seasonal evolution of the drainage system and to underline the importance of hydraulic isolation at the glacier bed.

Now, to explore the spatial structure of the drainage system and its seasonal progression, we automatically cluster boreholes based on similarities in their pressure records and follow their evolution through the melt season. Some of these borehole clusters show pressure variations that suggest they are part of a drainage system connected to the surface meltwater supply, while others show features consistent with hydraulic isolation. The distribution of connected and isolated boreholes suggests that the distributed drainage system we observe comprises a network of small conduits with spacings smaller than the borehole

bottom diameter (approximately 25–50 cm). Within these hydraulically connected areas, pressure phase lags, and amplitude attenuation rarely shows the behaviour expected in a diffusive system. This observation suggests that the diffusivity distribution in such areas presents a fine structure at scales smaller than our minimum borehole spacing of 15 m. However, at a glacier-wide scale, we observe that hydraulic connections are ubiquitous in some regions of the bed and permanently absent in others, suggesting large contrasts in difuisivity.

Within disconnected areas, boreholes often show small amplitude pressure variations associated with horizontal normal stress transfers. Such stress transfers seem to play a more important role than previously considered for controlling the effective pressure distribution at the bed.

Through the melt season, the evolution of borehole clusters suggests that the diurnal meltwater supply promotes the growth of the low-efficiency drainage systems found early in the season while stimulating the shrinkage and fragmentation of the more

efficient drainage systems that appear later in the season. Therefore, an increase in drainage efficiency is associated with the growth of disconnected areas.

Our observations support the traditional view of a distributed drainage system early in the melt season that gradually evolves into a progressively more channelized system. However, the most notable difference is the highly heterogeneous distribution of diffusivity that our results suggest and the robust support for disconnected areas. The extent of disconnected areas could





be an essential control of basal speed variations. It is possible that even relatively small disconnected areas could have a disproportionate effect on basal speed.

# 1 Introduction

Glacier speed and ice transport rates are strongly influenced by the basal processes through their ability to modulate basal
sliding rates. The contribution of basal sliding to overall ice transport is especially important for large fast-flowing glaciers. For example, in the largest outlet glacier of the Greenland ice sheet (Jakobshavn Isbræ), basal sliding has been found to account for 44% to 90% of the measured surface speed (Lüthi et al., 2002; Ryser et al., 2014b). Similarly, in Antarctic ice streams, basal sliding can account on average for about 69% of the observed surface speed (Engelhardt and Kamb, 1998). Basal sliding rates often show a marked seasonal variation, with summer sliding speeds two to three times faster than winter averages (Nienow
et al., 1998a; Sole et al., 2011; Ryser et al., 2014b). These variations are a consequence of changes in the subglacial drainage system associated with the seasonal input of surface meltwater (Iken and Bindschadler, 1986; Gordon et al., 1998; Nienow et al., 1998b; Mair et al., 2001; Harper et al., 2005). However, those changes in the subglacial drainage system are one of the least observed glaciological phenomena, and we have only a limited understanding of how they take place, which physical processes are involved, and how do they influence basal sliding rates.

The main variable linking subglacial drainage processes to basal sliding is the effective pressure, defined as the difference between normal stress and water pressure at the bed, where normal stress is usually taken to be equal to the overburden pressure (OBP). In turn, the OBP corresponds to the weight of the ice column. Other variables that play a role in modulating basal sliding, include the size and distribution of bedrock heterogeneities, the presence of basal till, and the size and abundance of rock clasts embedded in basal ice (Weertman, 1957; Alley et al., 1986; Alley, 1989). Although these factors can change
significantly from one glacier to another, they are unlikely to control basal speed variations at seasonal or shorter timescales at a given glacier. Therefore, we will concentrate our attention on the role of effective pressure.

When effective pressure is low, the corresponding high basal water pressure provides partial support for the weight of the glacier, and therefore enhance basal sliding (Lliboutry, 1958; Hodge, 1979; Iken and Bindschadler, 1986; Fowler, 1987; Schoof, 2005; Gagliardini et al., 2007). A similar effect is observed on glaciers resting on a till layer, where a lower effective
pressure reduces the yield stress of the till, and therefore also enhances basal sliding (Engelhardt et al., 1978; Iverson et al., 1999; Tulaczyk et al., 2000; Truffer et al., 2001). Conversely, large effective pressures enhance the mechanical coupling at the bed interface and therefore reduce sliding.

Many recent subglacial drainage models (Schoof, 2010; Hewitt, 2011; Schoof et al., 2012; Hewitt et al., 2012; Hewitt, 2013; Werder et al., 2013; Bueler and van Pelt, 2015; Downs et al., 2018; Sommers et al., 2018) consider a pervasive subglacial
drainage system that covers all of the ice-bed interface. Therefore, such a system can effectively transmit effective pressure variations across the entirety of the glacier bed. Drainage models of this type have succeeded in reproducing many of the observed variations of glacier velocities at a seasonal scale, and the seasonal up-glacier development of a channelized drainage



system during the spring and summer (Hewitt, 2013; Werder et al., 2013). However, these models still fail to reproduce direct borehole observations (Flowers, 2015).

In Rada and Schoof (2018), through the study of a large network of boreholes in a small alpine glacier, we showed that most of the borehole observations at odds with models predictions can be understood as the result of hydraulically isolated areas at the glacier bed. These areas are characterized by boreholes that show constant or slowly varying water pressure, while other nearby areas display diurnal pressure variations in response to the surface meltwater supply (Hodge, 1979; Engelhardt et al., 1978; Murray and Clarke, 1995; Gordon et al., 1998; Hoffman et al., 2016; Rada and Schoof, 2018). Other common borehole observations that can arise from isolated areas, but cannot be explained by a pervasive subglacial drainage system are:

1. Large and sustained pressure gradients over short distances (Murray and Clarke, 1995; Iken and Truffer, 1997; Fudge et al., 2008; Andrews et al., 2014).

2. The development of widespread areas of high water pressure during winter (Fudge et al., 2005; Harper et al., 2005; Ryser et al., 2014a; Wright et al., 2016).

3. Boreholes exhibiting persistent pressures that exceed the overburden pressure (Gordon et al., 1998; Kavanaugh and Clarke, 2000; Boulton et al., 2007).

4. Boreholes exhibiting mutually anti-correlated diurnal pressure variations (Murray and Clarke, 1995; Gordon et al., 1998; Andrews et al., 2014; Lefeuvre et al., 2015; Ryser et al., 2014a).

At South Glacier, more than 25% of the borehole observations fall under some of the above categories during summer and almost 100% during the winter, when most boreholes display pressures near or above overburden for several months.

To understand how all these observations can be explained by the existence of isolated conduits or "water pockets" at the bed, it is important to consider two key processes: The first is ice creep, which acting on the walls of an isolated water pocket will change its volume and internal water pressure until equilibrium is reached at a value close to overburden (Fig. 1a).

The second is the effect of horizontal normal stress transfers. These stress transfers can either reduce or increase the normal stress in some regions of the bed. Figure 1b illustrates how a decrease of the water pressure within an isolated water pocket can take place when the water pressure of a nearby connected conduit is higher than the normal stress in its surroundings. Such pressure excess would offer partial support of the overlying ice, thus reducing the normal stress around the water pocket and its internal water pressure. Murray and Clarke (1995) termed this process as "load transfer" (see also Weertman, 1972; Gordon et al., 1998; Lappegard et al., 2006; Lefeuvre et al., 2015). Conversely, if the water pressure within the connected conduit is lower than the normal stress, part of the unsupported weight of the ice above the conduit will be transferred to the surrounding bed and any nearby water pocket, increasing the water pressure within it (see Fig. 1c-d). This process is referred to as "bridging stress" by Lappegard et al. (2006). Mutually anti-correlated pressure variations can also be understood as the response of isolated water pockets forced to keep a fixed water volume during changes in the normal stress in the surrounding ice. Such changes in normal stress can be due to any of the previously described normal stress transfers (Fig. 1c-d).



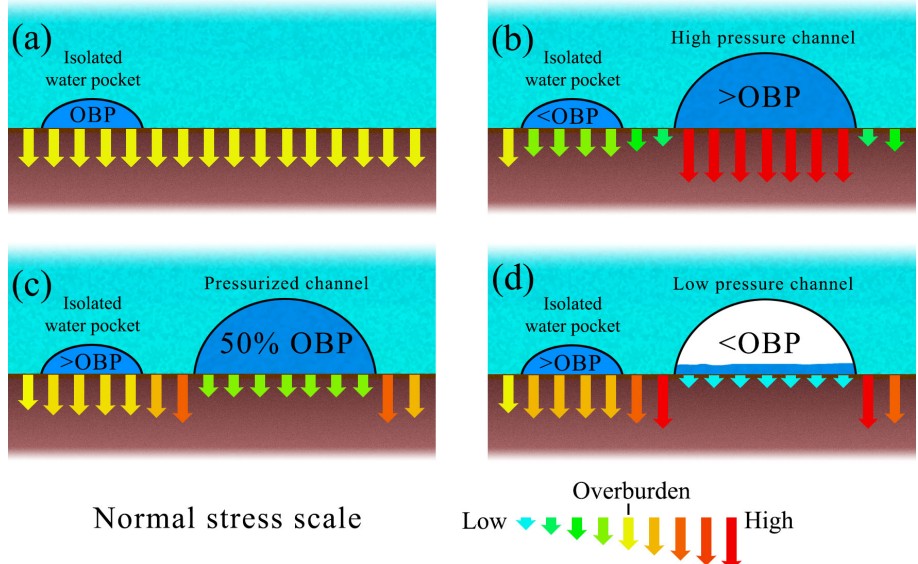

**Figure 1.** Effect of ice overburden and normal stress transfers on isolated water pockets. (a) In the absence of an active drainage system, the water pressure in an isolated water pocket reaches equilibrium close to overburden pressure. (b) A conduit with an internal water pressure above overburden reduces the normal stress in the surrounding bed leading to a pressure below overburden on nearby isolated water pockets. (c-d) A conduit with an internal water pressure below overburden increases the normal stress in the surrounding bed leading to above overburden pressures on isolated water pockets. Pressure variations in the channel would produce anti-correlated variations in the water pocket.

Another consequence of the existence of relatively large disconnected regions at the glacier bed, is the reduction of the area of influence of the active subglacial drainage system. Consequently, the extent of the disconnected regions of the bed could play an important role in controlling basal sliding and its sensitivity to changes in meltwater supply.

The identification of widespread areas in hydraulic isolation at South Glacier and other glaciers, motivates the need for a better understanding of the spatial structure of the subglacial drainage system and its evolution through time. In particular, how does the extent of connected and disconnected areas evolve, and how does that evolution relate to the seasonal cycle of meltwater supply. With this motivation, we build here upon the work presented in Rada and Schoof (2018), developing a methodology to identify connected and disconnected areas of the glacier bed, and how their distribution changes through the seasonal cycle.

## 1.1 Inferring subglacial hydraulic connections

Generally, we cannot observe directly the geometry of the subglacial drainage system, and have to rely on inferences made from borehole observations. The most common approach to this problem is to assess the efficiency of the connection between pairs of boreholes based on their response to a common forcing signal, which can be natural or artificial. While a process-based



approach might be preferable, such as the inversion of a forward model, doing so would require the forward model to account for the full phenomenology of the borehole records, and such a forward model does not yet exist.

During the spring and summer months, the subglacial drainage system is forced by a quasi-diurnal cycle in surface meltwater supply. The distinct response of each borehole to this forcing can be used to assess the efficiency of the connections between
them. A connection is efficient if the two boreholes display a similar response to the forcing, and inefficient otherwise. More specifically, a connection is efficient when the hydraulic conductivity of the conduit system connecting two boreholes is high and the water storage capacity of that system is low. Therefore, boreholes showing a very similar pattern of pressure variations are likely to be well connected, while boreholes showing a very different pattern are poorly or nor at all connected.

It is important to note that this approach to the identification of subglacial connections relies on the ability of each drainage
subsystem to modulate the forcing signal in a distinct way. That modulation is the result of the specific geometrical structure, permeability and storage capacity of each subsystem. However, differences in water pressure variations between subsystems could also arise from differences in the forcing, as melt water production can vary across the glacier surface. This variation can result from differences in albedo, slope or shadowing. Although we cannot distinguish between differences in observed pressure response that arise from internal properties or forcing changes, arguably, two distinct subsystems would still represent
areas that evolve with some degree of independence.

More problematic for the identification of subglacial hydraulic connections is the possibility that two distinct subsystems could display indistinguishable responses to the same forcing. A method based on the similarity of diurnal pressure response can erroneously aggregate mutually-disconnected areas of the drainage system into a single subsystem. However, the extent of the differences observed at South Glacier in the responses of neighbouring subsystems to meltwater supply suggests that
independent subsystems generally modulate the forcing signal to a point where they become well differentiated from each other (Rada and Schoof, 2018). This observation is also consistent with those reported from other glaciers (Fountain, 1994; Gordon et al., 1998; Harper et al., 1998; Fudge et al., 2008).

Subglacial hydraulic connections have also been studied using artificially induced signals. One approach is to use tracers such as salt or fluorescent dyes (Hubbard and Nienow, 1997). If the injected tracer is detected at a given location, the injection
site and that location must be hydraulically connected. However, hydraulic connections that are not associated with significant water exchange cannot be detected in this way, although they could be equally or more relevant to the control of the overall effective pressure at the bed. An alternative approach is the use of slug tests (Stone, 1993; Waddington and Clarke, 1995; Stone and Clarke, 1996; Iken et al., 1996; Kulessa et al., 2005). On glaciers, this method usually consists of studying the water level changes in an open borehole after an initial artificially induced level change. However, the logistical challenges associated with
performing repeated tracer injections or slug test year-round in multiple locations, have prevented them to be used in long-term studies of the subglacial drainage. In addition, slug tests may also actively alter the drainage system. In contrast, the relative simplicity and less invasive nature of continuous pressure measurements in boreholes have made them common practice for the study of subglacial hydraulic connections (Gordon et al., 1998; Harper et al., 2002; Fudge et al., 2008; Huzurbazar and Humphrey, 2008).





Based on the similarity of the response of boreholes to natural diurnal forcing, hydraulic connections between boreholes have been detected automatically using two different clustering techniques. Fudge et al. (2008) used K-means clustering (MacQueen, 1967) to group boreholes with similar responses to diurnal forcing at Bench Glacier, Alaska. Although this is a simple and effective clustering technique, it is hard to automate due to the requirement that the number of clusters within the dataset needs to be known a priori. This shortcoming was pointed out by Huzurbazar and Humphrey (2008), who instead used hierarchical clustering. In contrast to K-means, hierarchical clustering groups together all the sensors that conform to a given degree of similarity.

While the identification of these "clusters" of similarly behaving boreholes provides useful information about the structure of the subglacial drainage system, this is only a snapshot of the subglacial drainage system, and gives no insight into how it is evolving. As the subglacial drainage system changes continuously in response to the seasonal cycle, we need a sequence of these snapshots to capture the evolution of the system throughout the year. For this reason, we present here a new technique that allows the identification and follow-up of these clusters through time.

Using this new technique, we will study how the structure of the subglacial drainage evolves on a small alpine glacier, how the extent of connected and disconnected areas changes, and how the effective pressure varies within these two different regions of the bed. With this information, we will present a comprehensive picture of the seasonal evolution of the drainage system. This picture is broadly consistent with the standard description of an extensive early-season distributed drainage system, that progressively evolves into a channelized system during the summer. However, it also provides further evidence for the existence of extensive disconnected regions of the bed, and suggests the almost complete shut-down of the subglacial drainage over winter. It suggests as well that horizontal normal stress transfers play a more important role than previously considered for the control of the effective pressure distribution at the bed, and it will present a novel window into the fine structure of the subglacial diffusivity distribution.

## 2 Field site and methods

### 2.1 South Glacier field site

All observation presented were made on a small (4.28 km$^2$), unnamed surge-type alpine glacier in the St. Elias Mountains, Yukon Territory, Canada, located at 60° 49' N, 139° 8' W (Fig. 2). We will refer to the site as "South Glacier" for consistency with prior work (Paoli and Flowers, 2009; Flowers et al., 2011, 2014; Schoof et al., 2014). Surface elevation ranges from 1,960 to 2,930 m above sea level. Direct instrumentation and radar scattering (Wheler and Flowers, 2011; Wilson et al., 2013) reveal a polythermal structure with a basal layer of temperate ice overlaid by cold ice.

An automatic weather station (AWS) operated at 2,290 m next to the lower end of the study area (see Fig. 2) between July 2006 and August 2015 (MacDougall and Flowers, 2011) as part of a simultaneous energy balance study (Wheler and Flowers, 2011). We use air temperatures (specifically positive air temperatures, meaning the maximum of measured temperature and 0° C) and Positive Degree Days (PDD, defined in the usual way as the integral with respect to time over positive air temperatures) as the main proxy of the water input into the subglacial drainage system. Temperature estimates after the August 2015



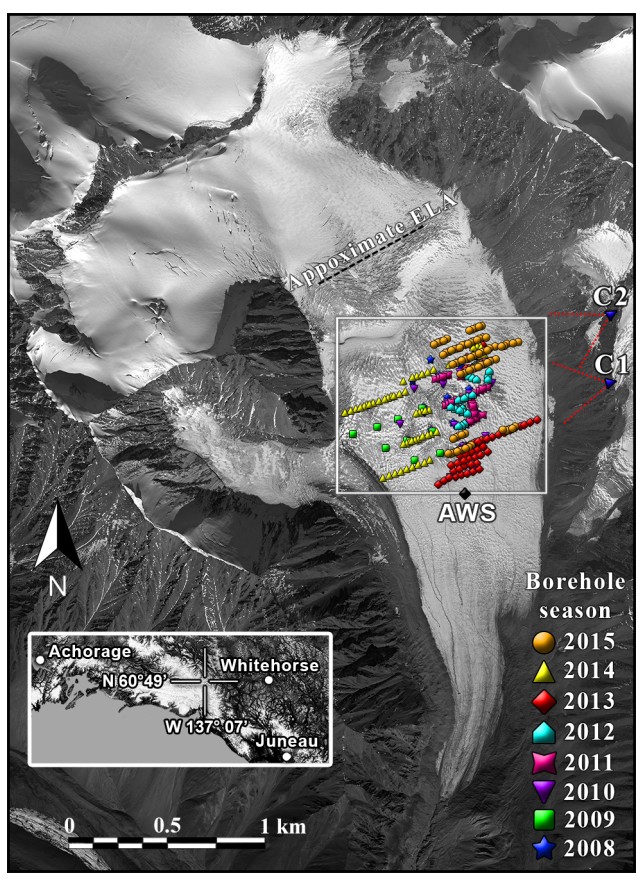

**Figure 2.** WorldView-1 satellite image of South Glacier taken on September 2nd, 2009. Borehole positions are marked according to the year of drilling, showing the most recent year in repeatedly drilled locations. Time-lapse camera positions (C1 & C2), Automatic Weather Station (AWS) approximate equilibrium line (ELA) are also indicated. The inset map shows the general location in the Yukon. The white box corresponds to the boundaries of the study area.

removal of the on-glacier AWS were calculated by a calibrated linear regression of data from a second AWS operated since 2006 by the Geological Survey of Canada and the University of Ottawa 8.8 km to the Southwest, at an elevation of 1845 m. The approximate extent of snow cover over the study area was assessed visually using time-lapse imagery.

Surface velocities were measured with a GPS array (Flowers et al., 2014), and display a strong seasonal contrast. The velocity near the center of the study area (white rectangle in Fig. 2) varied from 30.6 to 17.9 m/year between summer 2010 and early spring 2011. Modelled basal motion in our study area accounts for 75–100% of the total surface motion (see Fig. 6b in Flowers et al. (2011), where our study area is located between 1600 and 2500 metres).

Between 2008 and 2015, 311 boreholes were drilled to the bed (Schoof et al., 2014) in the upper ablation area of the glacier between 2,270 and 2,430 m asl (Fig. 2), covering an area of approximately 0.6 km$^2$, with an average ice thickness of 63.4 m and a maximum of 100 m. No moulins are visible in or above this area. Instead, the surface meltwater is routed into the

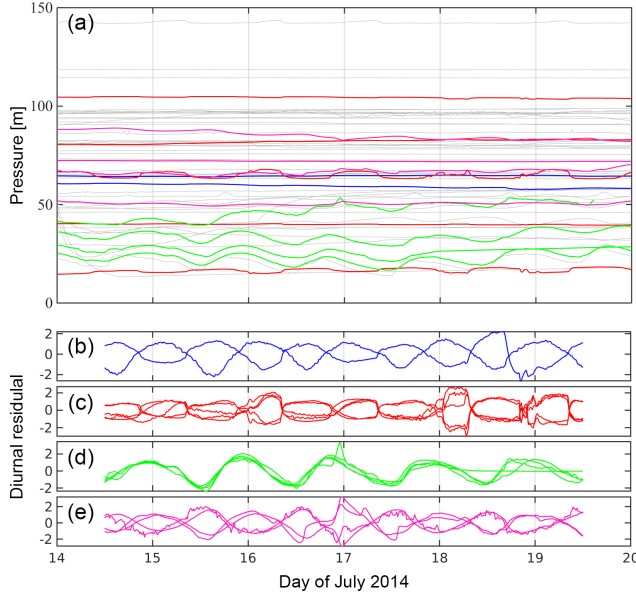

**Figure 3.** (a) Raw pressure records of 60 time series over a 6-day window starting on July 14th, 2014. Time series belonging to four manually identified clusters are shown with thick lines. (b) to (e) Diurnal residuals of the sensors belonging to each of the four identified clusters (same colour coding of panel a). Diurnal residuals are normalized by their standard deviation, resulting in a dimensionless quantity.

glacier through abundant crevasses. The basal layer of temperate ice in the study area extends up to 30–60 m above the bed. A relevant consequence of this polythermal structure is that the upper end of the boreholes typically freezes shut within few days. Boreholes were instrumented with pressure transducers providing continuous subglacial water pressure records, with up to 150 boreholes being recorded simultaneously. More details on the field site, drilling methodology, instruments used and data

quality assessment can be found in Rada and Schoof (2018).

## 2.2   Identification of subglacial drainage structures

To infer subglacial hydraulic connections, we will look for pressure time series that display similar diurnal variations. However, these variations are, in general, time-limited. For this reason, we will look for this similarities over discrete time windows. We will discuss in detail how the length of this time window is chosen. The exercise of identifying by eye which time series

display similar pressure variations over a given time window becomes onerous as the number of time series and the differences between them increase. Figure 3, shows 60 time series over a 6-day window. Among those time series, it is possible to identify some similarities. For example, the four green lines show very similar diurnal pressure variations, with similar amplitudes but a distinct pressure offset. In contrast, the red lines do not appear to be similar to each other, and some of them seem to be flat lines. In this case, the similarity is difficult to identify because the pressure offset between the boreholes is much larger than

the amplitude of diurnal pressure variations. For that reason, the identification of similarities can be substantially facilitated by the subtraction of the mean value from each time series. However, if the time series consist of diurnal variations superimposed





on a long-term trend, subtracting the mean value might not be sufficient, because the pressure range covered by the trend can also be large enough to render the diurnal variations imperceptible. Therefore, we subtract from each time series its running mean over a 1-day window. Mathematically, given a time series $P$ with samples $P_i$ at regular time intervals, we remove the running mean over a 1-day interval, defining a "diurnal residual" $R_i$ through

$$R_i = \frac{1}{\sigma_{window}} \left( P_i - \frac{1}{d} \sum_{j=i-d/2}^{i+d/2} P_i \right) \tag{1}$$

where $d$ is the number of samples contained in one day, and $\sigma_{window}$ is the standard deviation of the time series $P$ within the window over which the similarity comparison will be performed. The normalization by the factor $\sigma_{window}^{-1}$ facilitates the identification of similar time series regardless of the amplitude of their diurnal variations. This normalization is essential to reveal the similarities between hydraulically connected boreholes and those affected by normal stress transfers controlled by the former. It also allow us to identify the similarities between boreholes affected by other mechanical interactions. However, this normalization discards amplitude information that could be relevant in distinguishing between different drainage subsystems if they display a similar pattern of diurnal pressure variations. After the clustering process, we will incorporate this missing information to the analysis in order to identify the process responsible of the observed diurnal residual similarity.

We will term this diurnal residual transformation as "pre-processing", referring to the fact that it is applied to the raw data before attempting to identify similarities. Panels b to e of Fig. 3 show the diurnal residuals of the four groups of similar time series that we found among the 60 shown in panel a. In panel b we can see how the diurnal residual makes clear the similarity between the red lines and the same happens for the other groups.

The similarities between time series change in time as the structure of the subglacial drainage system evolve through the opening and closing of conduits. To capture this evolution, we break the data set into discrete time windows over which we will search for similar time series. Even with the aid of the diurnal residual pre-processing, the manual identification of similarities among hundreds of time series is time-consuming, difficult and prone to omissions, making it unsuitable for analyzing several hundreds of time windows with up to 150 time series each.

To overcome this limitation, we will use an automatic clustering method to define groups of "similarly-behaving" boreholes. The particular method and the parameters used in the algorithm are chosen to optimally reproduce sets of manually-picked borehole records that exhibit similar diurnal responses to surface melt. To identify these similarly-behaving boreholes systematically, we will look for groups of boreholes that display a similar pattern of the diurnal pressure variations represented by their diurnal residual. We will refer to those groups as "clusters". Then, for each cluster, we will try to identify the physical process causing the similarity. Following the work presented in Rada and Schoof (2018), we will distinguish two broad types of processes responsible for similarity: hydraulic connections, and mechanical interactions.

When we have evidence that a group of boreholes shares a common pattern of pressure variations as a consequence of mechanical interactions only, we will refer to it as a "mechanical cluster". Otherwise, we will refer to it as a "hydraulic cluster". A disconnected borehole that displays the same pattern of pressure variations as a hydraulic cluster but in inverted form (presumably due to a normal stress transfer) will also be included in the hydraulic cluster, although our clustering method





will be able to distinguish connected and disconnected boreholes within the cluster. These disconnected boreholes together with their hydraulically connected counterparts will define an area of influence that extends beyond the reach of the hydraulically connected part of the cluster.

To find the most suitable technique to identify borehole clusters in a large dataset such as the one available at South Glacier,
we have tested four different clustering methods: K-means (MacQueen, 1967), Hierarchical clustering (Rokach and Maimon, 2005), Self-Organizing Maps (SOM) (Vesanto et al., 2000), and Empirical Orthogonal Functions (EOF) (Jolliffe, 2002). We tested the capacity of each method to reproduce automatically a set of clusters picked by hand, finding that hierarchical clustering was the best of the four clustering techniques for our application (see supplementary information).

Therefore, the automated clustering process consists of the following steps:

1. We subdivide the data in discrete overlapping time windows.

2. In each window, we find all the available time series, which are then interpolated to regular time stamps with 15 minute spacing. Data gaps up to 30 minutes were linearly interpolated. Longer data gaps resulted in the exclusion of the time series from the corresponding time window.

3. Computation of the diurnal residual of each time series.

4. Application of the agglomerative hierarchical clustering method.

The agglomerative hierarchical clustering method (Rokach and Maimon, 2005) is an iterative clustering technique that starts from a set of one-element clusters (single pressure residual time series), and then, in each iteration, merges the pair of time series that display the higher degree of similarity into a larger cluster. This process effectively organizes all the original time series into a tree-like structure termed a "dendrogram". To identify the two most similar time series in each iteration, we need to
quantify what we mean by similarity between two time series. This is done using a metric that defines a generalized "distance" between two time series. The smaller the distance the more similar the two time series are. We will define the metric we use shortly, but first we will illustrate the clustering process graphically.

Figure 4 shows a dendrogram computed for the 60 time series presented in Fig. 3a. Lines in the dendrogram are termed "branches", and the joints between branches are "nodes". The vertical position of a node represents the distance between its
lower branches. Therefore, similar clusters join lower in the dendrogram than dissimilar ones. To find the clusters of time series conforming to a given degree of similarity, we define a Split Point (SP). The SP establishes the maximum distance allowed between time series that belong to a single cluster. Once the SP is defined, we select the clusters forming below it as candidates for a hydraulic or a mechanical clusters. As an example, the coloured branches in Fig. 4 represent clusters that would be selected using the SP defined by the dotted black line. Those clusters correspond to the time series shown on panels b–e of
Fig. 3. It is important to note that we have also tested a scheme in which we define a separate SP for hydraulic and mechanical clusters. However, as the SP values found for each type are very similar, we have preferred the use of a single SP for both types of clusters.

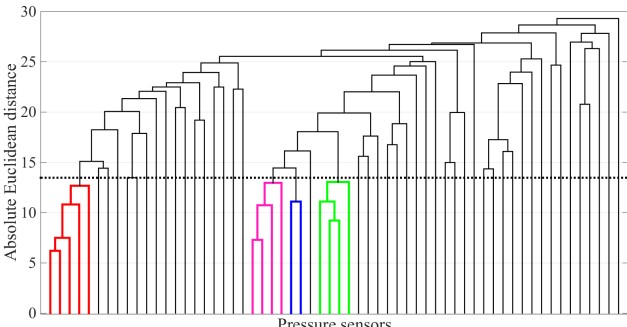

**Figure 4.** Example of a dendrogram computed by agglomerative hierarchical clustering over the 60 time series presented in panel a of Fig. 3. The coloured lines correspond to the four clusters shown in panels b–e of Fig. 3. The thick dotted line corresponds to a Split Point (SP) that would output the same four identified clusters. Diurnal residuals are normalized by their standard deviation, therefore, their values and their euclidean distances are dimensionless quantities.

The size of the time window over which the clusters are identified has a considerable impact in the resulting clustering. For our purposes, a useful window size must be longer than the main one-day period of the pressure variations, but shorter than the time required for significant changes in the subglacial drainage to take place. This criterion loosely constrains the window size from a few days to a few weeks, where the upper limit is fairly speculative. However, the observed changes in the
pressure records suggest that the drainage system can undergo significant changes within two weeks, making that time scale a reasonable upper limit. Within that range, longer windows can better discriminate between different subsystems, and shorter ones can resolve more stages in the evolution of the subglacial drainage. We use a time window of six days, that aims to strike a balance between sensitivity and temporal resolution: it is long enough to capture multiple diurnal cycles and the length of a typical weather system in the area, but at the same time it is short enough to provide a detailed sequence of the evolution of the
subglacial drainage.

Our aim with the clustering is first to find all the boreholes showing similar diurnal residuals, and later discriminate which physical process is responsible for their similarity. In particular, we want to establish whether the similar time series are consistent with the existence of a hydraulical connection or a mechanical interaction. In the case of mechanical interactions such as normal pressure transfer (Murray and Clarke, 1995; Gordon et al., 1998; Lappegard et al., 2006; Lefeuvre et al., 2015),
basal slip events (Andrews et al., 2014), or bridging stresses (Weertman, 1972; Lappegard et al., 2006), the similarity between pressure records is limited to the relative pattern of pressure variations, while they can differ widely in their absolute value, amplitude and long-term trend. It is important to note that differences in absolute values and long-term trend are removed by the diurnal residual pre-processing, allowing them to be clustered together. Mechanical interactions can also invert the direction of the variations, with peaks becoming troughs and vice versa.
Motivated by processes that can invert the pattern of pressure variations, we choose a distance metric insensitive to that form of inversion. Therefore, we use an "absolute Euclidean distance": given two time series $A$ and $B$, with samples $a_i$ and $b_i$





respectively and with $i = 1, ... N$, we define the absolute Euclidean distance between $A$ and $B$ as

$$D(A,B) = \min\left( \sqrt{\frac{1}{N}\sum_{i=1}^{N}(a_i - b_i)^2}, \sqrt{\frac{1}{N}\sum_{i=1}^{N}(a_i + b_i)^2} \right) \qquad (2)$$

This corresponds to the minimum of the Euclidean distance between $A$ and $B$, and between $A$ and $-B$. Therefore, the absolute Euclidean distance will assign small distances to pairs of similar time series, even if one of them is an inverted

version of the other. When operating over standardized time series (i.e. normalized by the standard deviation) as in our case, the Euclidean distance is mathematically equivalent to the correlation coefficient. Previous work in subglacial hydrology has used Euclidean distance for clustering, either directly on the pressure time series (Fudge et al., 2008) or its first derivative (Huzurbazar and Humphrey, 2008).

The absolute Euclidean distance as described above applies only to individual time series. However, hierarchical clustering

requires the calculation of the distance between clusters of time series. The method used for such calculations is known as the "linkage". We use the average-link linkage (Rokach and Maimon, 2005), where the distance between two clusters corresponds to the average distance between the time series in one cluster and the ones in the other.

Section "Clustering calibration, validation, and testing" of the supplementary material provides detailed information on the criteria we used to identify similar time series and how we calibrated, validated and tested the methodology used here to

optimally reproduce manually picked clusters.

## 2.3 Cluster evolution in time

To study the evolution of the drainage subsystems, we apply the calibrated hierarchical clustering method to the whole dataset over a moving window of 6 days, with neighbouring windows overlapping by 3 days. After independently clustering successive time windows, we apply a custom algorithm to identify whether a cluster identified in one window is newly formed or

corresponds to a pre-existent cluster already identified in previous windows. Without such a "tracking" algorithm, it becomes challenging to follow the evolution of a particular area or set of boreholes. Also, continuity between successive windows is required to study the evolution of parameters such as the mean diurnal pressure, amplitude of pressure oscillations or the spatial extent of a given cluster.

Determining whether a cluster is new or constitutes the continuation of an existing one is somewhat ambiguous: if a cluster

splits into two clusters of equal size, it is unclear which branch to follow when we want to describe the evolution of the properties of the original cluster.

We have adopted an iterative approach: in the first iteration we consider that a given cluster continues in the following window as the cluster that shares the most sensors with it, and we resolve arbitrarily the ambiguities that arise when two successor clusters share the same number of sensors with the original one. This first iteration step successfully links clusters

but tends to create many short-lived clusters, instead of an equally consistent but more continuous sequence. For this reason, in the subsequent iterations we again choose from all the possible successors using the same criterion, but this time we look further into the following windows (that are now preliminarily linked), considering not only how many boreholes a cluster





shares with a potential successor, but also with the successor of the successor and so on, through a total of four windows. When counting the number of shared boreholes, we give different weights to each consecutive window: from the closest to the furthest, these weights are 0.5, 0.375, 0.25 and 0.125. After a few iterations, the cluster structure converges to a more continuous sequence.

## 2.4 Hydraulic and mechanical cluster types

Clusters with similar pressure records may arise from different physical processes. In particular, they can be the result of hydraulic connections between boreholes or due to a common response of isolated boreholes to stress changes in the ice (Rada and Schoof, 2018). These "mechanical clusters" look very different than hydraulic ones and are easy to tell apart by eye. Nevertheless, we have automated their identification using the Time Series Shapelets method (Ye and Keogh, 2009). This method allows us to take advantage of the characteristic shape of the diurnal cycle observed in mechanical clusters, especially during the melt season. The Time Series Shapelet method takes a dataset with time series belonging to multiple classes, in this case, Mechanical (M) and Hydraulic (H), and searches through all the sub-sections of a prescribed length $L$ within all time series. Each sub-section is termed a "shapelet", and the method tests the capacity of each shapelet to determine whether a given time series belongs to the class M or H. This is based on the minimum absolute Euclidean distance found between the shapelet and all the sub-sections of length $L$ within the given time series.

Using a calibration dataset that contains 49 mechanical and 156 hydraulic manually identified clusters, we applied the Time Series Shapelets method with $L = 1$ day to find the best shapelet to discriminate between the two classes. The best shapelet found is shown in Fig. 5. We use this shapelet to classify time series automatically as mechanical if their minimum absolute Euclidean distance (see Eq. 2) to the shapelet of Fig. 5 is smaller than 12.9. This value corresponds to the optimal threshold found for the discrimination between mechanical and hydraulic clusters within the calibration dataset. More details regarding the derivation of this threshold can be found in the supplementary material. Note that a shapelet is always a section of a single time series. Therefore, the shapelet shown in Fig. 5 corresponds to a 1-day long piece of the pressure record observed at one of our boreholes.

We label as hydraulic any cluster not classified as mechanical. Within most hydraulic and mechanical clusters, we can identify two subclusters, where the peaks of one correspond to troughs of the other and vice versa. In the diurnal residuals, the two subclusters show up clearly as inverted versions of each other. Figure 6 shows a clear example of a cluster involving 50 sensors, with one subcluster shown in red and the other in blue. Note that these two subclusters would have become independent clusters if the initial hierarchical clustering had been done using ordinary instead of absolute Euclidean distances.

We separate the two subclusters by computing the matrix of correlation coefficients between all members of the clusters. Then, all positive values are set to 1 and negative values to -1, effectively turning each row of the matrix into a sequence of values that, for one particular borehole, indicate which boreholes are correlated or anti-correlated with it. Then, these sequences are separated into two subclusters using K-means clustering (David and Vassilvitskii, 2007), allowing us to achieve the separation shown in Fig. 6 without manual intervention.

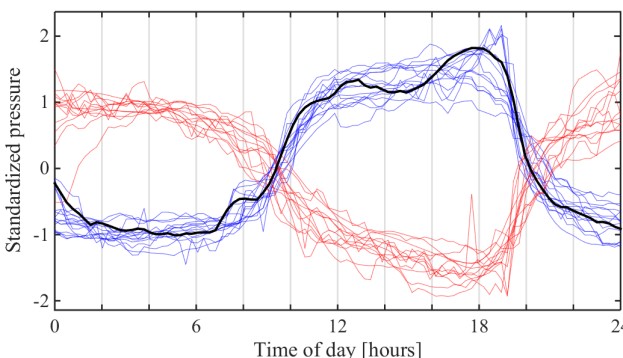

**Figure 5.** Best shapelet found for classification of mechanical connections (black line). This shaplet reached a 81% information gain (OSP = 12.9). For comparison, 23 time series of mechanical diurnal oscillations are also shown (blue and red lines).

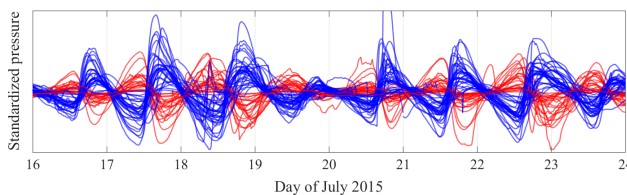

**Figure 6.** Diurnal residuals during 8 days for 50 sensors belonging to a hydraulic cluster. The two subclusters are presented in red an blue.

Figure 7 shows in panel b the standard deviation of the diurnal residual and the mean pressure of all the member time series of a hydraulic cluster that was tracked over 102 days. Panel a similarly shows a mechanical cluster that was tracked over 135 days. These clusters correspond respectively to the largest hydraulic and mechanical cluster observed during the 2015 melt season. To facilitate future references to these clusters, we will refer to them as "H1" and "M1" respectively. The values for

each borehole in Fig. 7 were computed using all the pressure records at that borehole during the periods of time where it was identified as a member of the cluster, and the standard deviation of the diurnal residual is provided as a proxy of the amplitude of diurnal variations. As in Fig. 6, one subcluster is shown in blue, and the other in red. We can see that there is a clear segmentation between the two subclusters in panel b, one having large amplitudes and high mean effective pressures (in blue), and the second having small amplitudes and low mean effective pressure (in red).

We interpret this as follows: large amplitudes and lower water pressures (higher effective pressure) are more likely to be associated with an active drainage system that drains surface meltwater, while low-amplitude pressure variations around overburden are likely to be the result of horizontal normal stress transfers (Murray and Clarke, 1995; Gordon et al., 1998; Lappegard et al., 2006; Lefeuvre et al., 2015). We will label the subclusters as correlated and anti-correlated, alluding to fact that boreholes in the correlated subcluster display maximum water pressures late in the afternoon, when the peak in meltwater supply

is expected. We have extended this labelling also to mechanical clusters, where correlation or anti-correlation is determined based on which subcluster peaks at the time period when we expect the maximum meltwater supply.





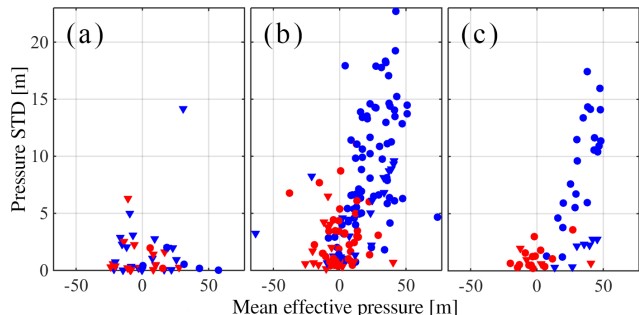

**Figure 7.** Scatter plots of mean effective pressure and pressure standard deviation for (a) mechanical cluster M1, (b) hydraulic cluster H1, and (c) window f of hydraulic cluster H1. Each point represents a borehole within the cluster. Boreholes that display diurnal variations in-phase with each other (i.e. belong to the same subcluster) are shown in the same colour. Boreholes are plotted as circles if located in the northern half of the study area or as triangles otherwise.

While the plots in panels a and b of Fig. 7 are useful to identify automatically the connected and unconnected subclusters, they do not offer an accurate representation of the real variation in mean effective pressure and amplitude. This misrepresentation is due to the differences in data availability and the length of time for which each sensor was part of the cluster. For example, if one sensor is part of the cluster only in a period where all sensors show small amplitudes, it will show up with an anomalously small amplitude. A representative example of the typical distribution of mean effective pressure and pressure standard deviation in the cluster H1 can be observed in Fig. 7c, where we display data only for one window of 6 days, from July 15th to July 21st, 2015. This corresponds to window f in Figs. 10 and 11.

Clusters were automatically identified in each window, tracked between windows, classified as mechanical or hydraulic, and divided into correlated and anti-correlated members. Subsequently, we performed a manual check of the automated output to correct apparent artifacts in the clustering process and handle exceptions like boreholes switching from correlated to anti-correlated (see Fig. 12), or clusters that switch from hydraulic to mechanical (see Fig. 13) or vice versa.

## 2.5 Spatial patterns in basal hydraulic connectivity

One of the questions we want to answer is whether the hydraulic properties of the ice-bed interface at South Glacier are homogeneous or if some areas are more likely to develop hydraulic connections than others. To address this question, we can study the spatial distribution of all the inferred hydraulic connections in our clustering output. However, comparing the changes in connectivity between different areas requires us to account for the spatial and temporal sampling biases in our dataset.

The spatial sampling bias arises from the fact that short-distance connections are more likely than long-distance ones. Therefore, a borehole will be more likely to make connections if it has many boreholes nearby than if it is relatively isolated. Similarly, the temporal sampling bias arises from the uneven data availability. Therefore, a borehole with a long pressure records will capture better the typical probability of connections than another with data limited to a few weeks, especially if the limited data covers a period of exceptionally high or low overall connectivity.



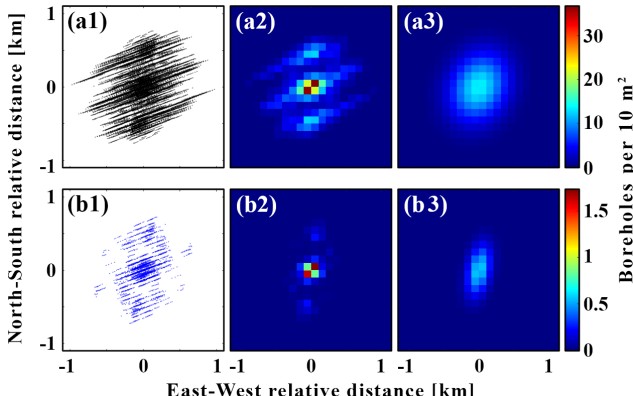

**Figure 8.** (a1) Relative positions of all 718,341 possible pairs of boreholes in selected time windows. (a2) Gridded density map of the relative positions in panel a1. (a3) Density map derived from the best bivariate Gaussian distribution fit to the relative positions in panel a1. (b1) Relative positions of all 9,514 pairs of boreholes identified as hydraulically connected in selected time windows. (b2) Gridded density map of the relative positions on panel b1. (b3) Density map derived from the best bivariate Gaussian distribution fit to the relative positions in panel a1.

To overcome these sampling biases, we will first assume that the bed is homogeneous and, under that assumption, we will estimate the probability of a hydraulic connection between two arbitrary points of the bed based on their relative position (i.e. distance and direction between them). Later, we will be able to test how well this probability can explain our observations, and assess the validity of the homogeneity assumption.

Note that we consider a hydraulic connection to have been identified between two boreholes when both boreholes are correlated members of the same hydraulic cluster over a given time window.

To estimate the probability of a connection across the bed, we consider how many hydraulic connections we have identified at a given relative position, and then estimate the probability of those connections based on how many times we have sampled for connections at such a relative position. In this calculation we use only the part of the year where we observe activity within

the drainage system. We achieve this by only using time windows for which we have identified at least one hydraulic cluster. Therefore, we ignore the extended winter period where we attribute the lack of connections to the absence of meltwater supply. We also assume that the connection probability can be represented by a bivariate Gaussian Probability Density Function (PDF).

We estimate this probability as

$$P(r,\theta) = \frac{D_{conn}(r,\theta)}{D_{boreholes}(r,\theta)} \tag{3}$$

where $r$ is distance and $\theta$ is the azimuth. $D_{conn}$ is a bivariate Gaussian PDF fit to the relative positions of all 9,514 pairs of boreholes for which we identified a hydraulic connection in the selected time windows. This PDF represents how likely a borehole in our dataset was to establish a connection with other borehole at distance $r$ and azimuth $\theta$. Figure 8b1 show all these relative positions, Fig. 8b2 shows a density map of the same positions gridded into 100 m by 100 m grid cells, and Fig.





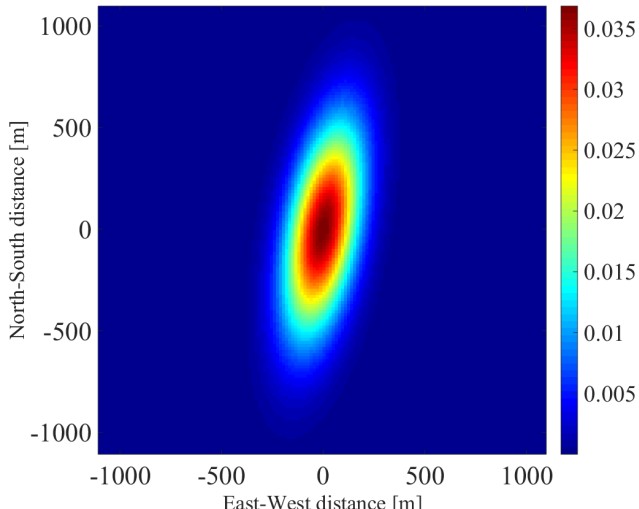

**Figure 9.** Probability density function for hydraulic connections $P$ as defined in eq. 3.

8b3 the density map expected from the bivariate Gaussian PDF fit for the same number of observations. Finally, $D_{boreholes}$ is a bivariate Gaussian PDF fit to the relative positions of all the 718,341 connections that would have been possible in all selected time windows. Therefore, $D_{boreholes}$ represents how likely a borehole in our dataset was to find another borehole at distance $r$ and azimuth $\theta$. Figure 8a1 shows all the relative positions, Fig. 8a2 shows a density map of the same positions, and Fig. 8a3

the density map expected for the same number of observations using the bivariate Gaussian PDF fit.

    Figure 9 shows the probability density function for hydraulic connections $P$ as defined in Eq. 3. This function will be used to estimate the number of connections we would have expected at a given borehole. That number of expected connections, corresponds to the sum of expected connections on each window in which that borehole contained a functioning pressure sensor. In turn, the number of expected connections for a given window corresponds to the sum of the probability of connection

with each one of the other boreholes recorded during that window. For example, consider a borehole that had a functioning pressure sensor in two time windows. In the first time window there were two other functioning boreholes and the probability of connection with them was 0.3 and 0.2. In the second time window, there were three other functioning boreholes with probabilities of connection of 0.1, 0.6 and 0.2. In this case, the expected number of connections would be the sum of all these probabilities, this is: 1.4 connections.

Differences between the expected and observed number of hydraulic connections at each borehole will be used to characterize different regions of the bed and assess the validity of the assumption of homogeneity implicit in our definition of the connection probability $P$.





## 2.6 Pressure variation trends

The study of the average pressure variation in multiple connected boreholes will be a useful tool to understand the evolution of the water pressure within the subglacial drainage system. However, due to the many discontinuities in our pressure records, and the wide range of mean values observed, the study of a simple average of the pressure records would not be very informative.

For example, if the data from a borehole with relatively high water pressure becomes unavailable at some point in time, the average pressure at that point would suffer a sudden drop. This pressure drop would be unrelated to any physical pressure change within the subglacial drainage system, and it would obscure the actual trend we are interested in.

Therefore, we will calculate the mean pressure of a series of boreholes by averaging the instantaneous pressure differences between consecutive samples of each borehole. Those averaged differences are then integrated in time to reconstruct a relative

averaged pressure time series for the whole interval. This relative averaged pressure starts at zero but represents accurately the pressure variations within the boreholes. As a final step, we add a constant value to the relative averaged pressure so that the mean value of it matches the mean of all the original pressure samples. To put this in mathematical terms, consider that each borehole is represented by a pressure time series with samples at times $t_i$, such that $P_{m,i}$ is the pressure recorded in borehole $m$ at time $t_i$. At each time $t_i$, the number of valid samples is $M_i$, such that boreholes can be represented by the index $m = 1...M_i$.

Then, the relative averaged relative pressure $R_i$ of all time series at time $t_i$ is

$$R_i = \sum_{k=2}^{i} \frac{1}{M_i} \sum_{m=1}^{M_i} P_{m,k} - P_{m,k-1} \tag{4}$$

then, the final averaged pressure time series $P_i'$ is given by

$$P_i' = R_i - \overline{R_i} + \overline{P_{m,i}} \tag{5}$$

where $\overline{P_{m,i}}$ is the mean of all samples in all time series and $\overline{R_i}$ is the mean value of the relative average time series defined by

$R_i$. All pressure time series in this chapter that show the average pressure variation of more than one borehole, were computed with the averaging defined by Eq. 5.

## 3 Results

### 3.1 Evolution of the subglacial drainage system

Data from the 2015 melt season represents our best record of the onset and evolution of subglacial drainage of South Glacier.

While we performed the clustering on the whole dataset, we will concentrate here on the 2015 melt season, where we had up to 157 working sensors, with about half of them (74) installed in previous years. The latter group produced a detailed record of the spring event and drainage development during the early season. Also, the 2015 melt season was long and warm enough
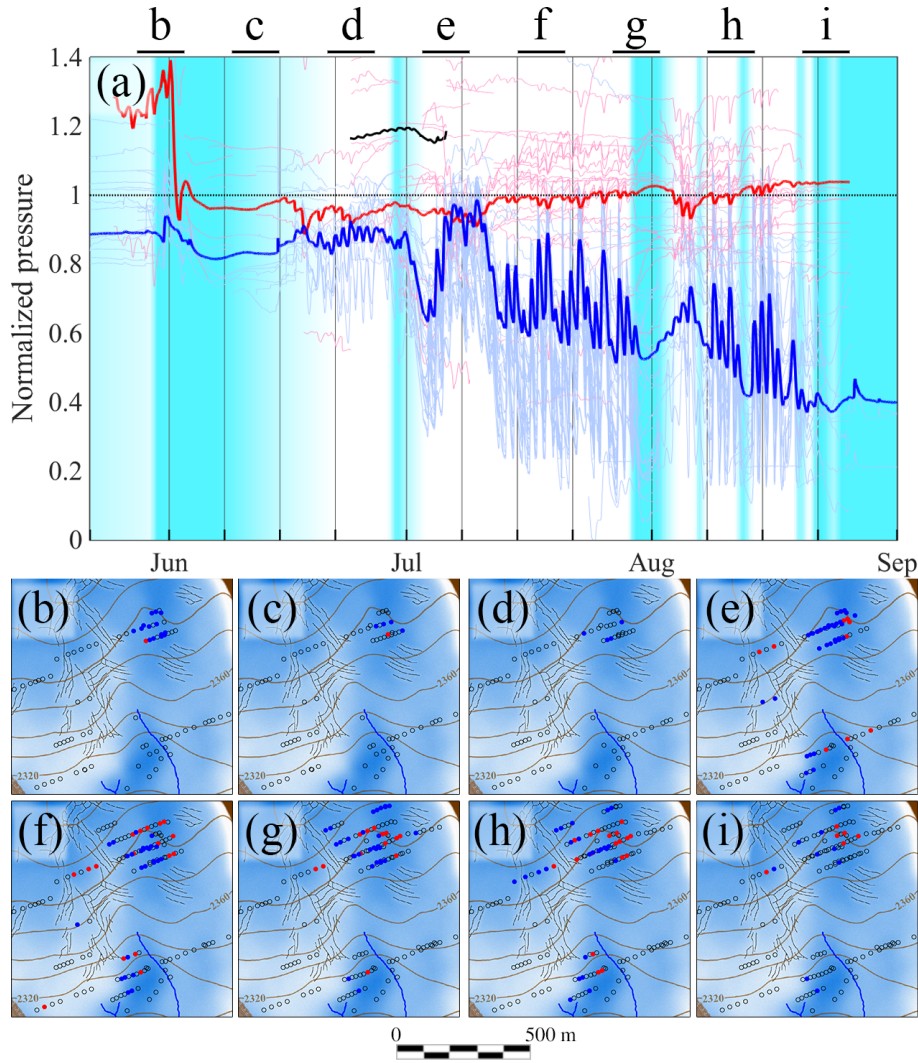

**Figure 10.** (a) Water pressure as a fraction of overburden for all correlated (blue) and anti-correlated (red) sensors participating in cluster H1. Thick lines represent mean values. The black line highlights a high-pressure correlated sensor, and the light blue shading the fraction of the glacier covered by fresh snow. Panels b to i show snapshots of the spatial distribution of correlated (blue circles) and anti-correlated (red circles) boreholes in eight time windows. Empty circles represent other boreholes that were recording pressure at the moment. The extent of the time windows associated with each snapshot is shown by the black bars at the top of panel a. The blue shading represents ice thickness.

to allow the formation of a well developed subglacial drainage system, something that does not occur every year. Nonetheless, results from the previous seasons are consistent with the observations of 2015.

To illustrate the evolution of a cluster during 2015, Fig. 10 shows the changes in mean pressure (panel a) and spatial distribution (panels b–i) for the correlated and anti-correlated boreholes of cluster H1. We can see how the mean pressure





within the correlated portion of this long-lived cluster steadily drops during the season, only punctuated by limited increases during periods of enhanced meltwater supply observed after two snow events around June 30th and July 31st. This decreasing trend in pressure through the season has also been observed by Gordon et al. (1998) at Haut Glacier d'Arolla.

While individual correlated boreholes share a common long-term pressure trend, anti-correlated boreholes display a wide

variety of long-term trends, the most common consisting of a constant pressure value. The mean pressure of the anti-correlated boreholes does not show a significant trend. Nevertheless, we observe an small increase of the mean pressure in anti-correlated boreholes over the season. While we are uncertain of the statistical significance of such trend, it would be consistent with the drop in mean pressure within connected (correlated) boreholes. Product of such pressure drop, the surrounding bed – were anti-correlated boreholes are located – must bear the normal stress left unsupported by connected areas.

The study of the evolution of individual clusters can only provide a limited picture of the overall dynamics. This overall picture includes the split of larger clusters into smaller ones, the merging of multiple clusters, or the appearance of numerous short-lived clusters. To visualize this processes, Fig. 11 organizes each cluster in a temporal network, where each small coloured box represents one of the clusters identified in a given window throughout the 2015 melt season. Clusters identified in the same time window are aligned vertically, and horizontally aligned series of boxes correspond to the different stages of one

individual cluster trough time. The time windows used during the clustering process were 6 days long, and neighbouring windows had a 50% overlap. However, for visualization purposes each box in Fig. 11 only covers two days around the centre of the corresponding window. The position of each cluster along the vertical axis has no physical meaning and has been chosen to improve visualization.

The height of each box is proportional to the number of boreholes within a cluster. However, changes in sampling through

the season as new boreholes were drilled and old sensors stopped working, can give a misleading idea of evolution. This effect can be seen in Fig. 10, where the growth of the cluster H1 between windows e and f is mostly associated with the incorporation of a new line of recently drilled boreholes. To properly account for sampling effects on cluster sizes, in Fig. 11 we scaled the height of each box by two factors. The first is the number of boreholes that are part of the cluster in a given window divided by the total number of working sensors between May and November 2015. The second is the ratio of the total number of boreholes

that formed part of the cluster for any part of 2015, to the number of working sensors in that window. The first factor scales clusters according to their relative size, and the second adjusts the scaling for the changing number of working sensors through the season.

We can see that the evolution of hydraulic clusters shows a quick onset, and rapid growth during periods of increasing meltwater supply. Panel c shows the PDD record, which is a good proxy for the rate of surface meltwater production (see

section 2 of the supplementary material of Rada and Schoof (2018)), and also a good proxy for the rate of meltwater supply to the subglacial drainage system over periods without fresh snow cover (white background in Fig. 11). During the first week of July, a substantial increase in meltwater supply led to the formation of an extended cluster (labelled H1) that incorporated all the connected sections of the bed.

We observe cluster growth mainly during the spring event, and to a lesser extent after snow events followed by high tem-

peratures later in the season, as illustrated by the cluster H1 after the snow event observed at the end of July 2015. When



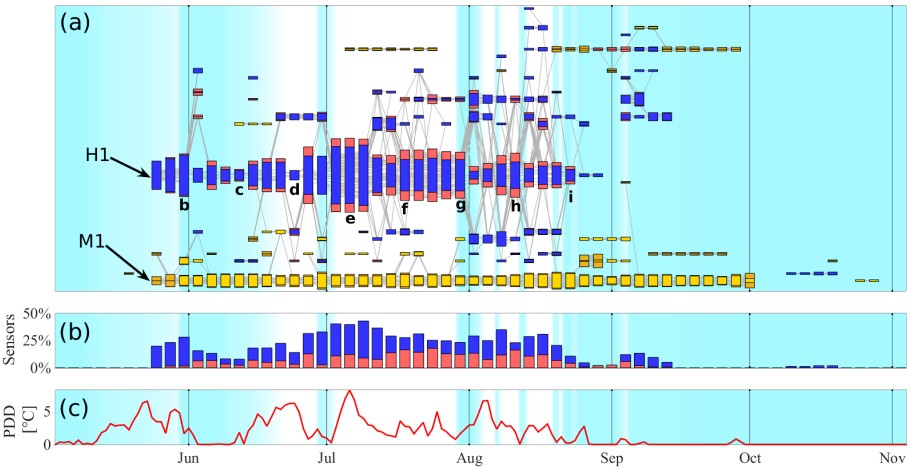

**Figure 11.** Cluster network for the melt season of 2015. In panel a, each sequence of aligned coloured boxes represents snapshots of a cluster trough time. Boxes labelled b–i correspond to the maps in Fig. 10. Hydraulic clusters are presented by blue and red boxes, where blue and red represents the fraction of correlated and anti-correlated boreholes respectively. Mechanical clusters are presented in shades of yellow. Thin grey lines represent the trajectories of individual boreholes. In the background, the light blue shading provides a qualitative representation of the fraction of the glacier covered by fresh snow as derived from visual inspection of time-lapse imagery. Panel b shows the total fraction of correlated (blue) and anti-correlated (red) sensors per window participating in hydraulic clusters. Panel c shows the daily PDD record.

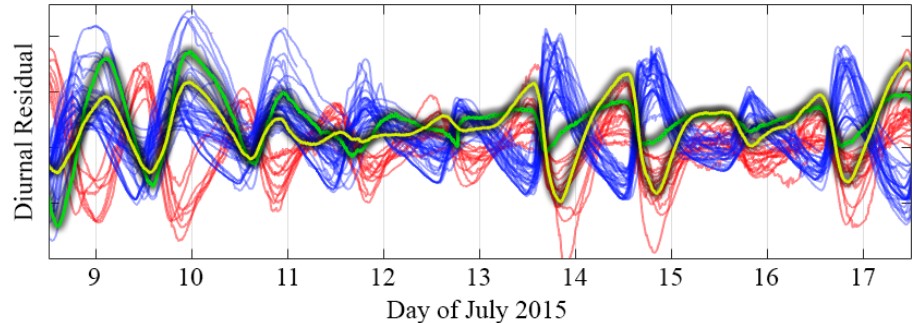

**Figure 12.** Diurnal residual data between July 9th and 17th 2015 for a hydraulic cluster. Over this period the cluster consisted of 41 correlated boreholes (blue), and 16 anti-correlated boreholes (red). Two additional boreholes (green and yellow lines), transition from correlated to anti-correlated around July 12-13th.

diurnally-averaged meltwater supply is steady or decreasing, hydraulic clusters experience a progressive reduction and fragmentation. We can observe this process in the evolution of cluster H1 during July 2015, and again during the second half of August 2015. The observed cluster size reduction happens by borehole disconnection and cluster fragmentation.

Boreholes that cease to be hydraulically connected to a cluster can connect to another hydraulic cluster or become entirely disconnected. In some cases, disconnected boreholes can turn into anti-correlated members in the same cluster, as is the case





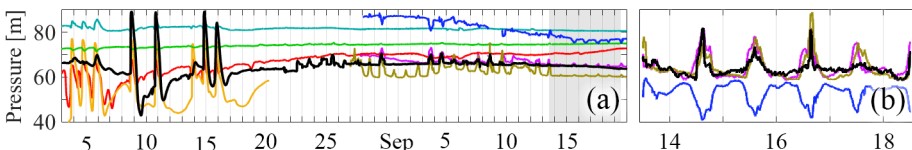

**Figure 13.** (a) Water pressure in a hydraulic cluster observed in August 2015 that we then identified as a mechanical in mid-September 2015. (b) Diurnal residual for the period between September 14th and 18th 2015, using the same colour coding of panel a. It can be seen how the sensor in black transitioned from displaying a large amplitude hydraulic signal to small amplitude one, characteristic of mechanical clusters.

for the two sensors shown in Fig. 12. In other cases, they can turn into members of a mechanical cluster, as illustrated in Fig. 13.

In the snapshots of the spatial distribution of cluster H1 shown in Fig. 10b–i, we can see that the growth of cluster observed in Fig. 11 is not only explained by the incorporation of new boreholes within the initial area of influence of the cluster, but also
by the growth in the spatial extent of the area of influence across the glacier. We can see how anti-correlated boreholes tend to show up preferentially on the edges of the connected regions, but they can also occur as "islands" within areas of the bed predominantly well connected to the subglacial drainage system (see Fig. 10e–h).

The fraction of correlated and anti-correlated boreholes in each window of cluster H1 is represented in Fig. 11 in blue and red respectively. Note that after the cluster H1 reached its peak size during the first week of July, the fraction of anti-correlated
boreholes increases while the cluster reduces its size (Fig. 11b). The fraction of anti-correlated boreholes for windows b–i of cluster H1 are 5%, 20%, 0%, 23%, 37%, 32%, 40% and 33% respectively.

Recall that we identify correlated boreholes by the larger amplitude of their diurnal variations, and by their mean pressures being generally lower than those of anti-correlated boreholes. Figure 7 shows the mean pressure and the amplitude of diurnal variations for all the boreholes in clusters M1 (panel a) and H1 (panel b).
There are exceptions to this general pattern. Some anti-correlated pressure records can display large amplitude oscillations, and most notably, some correlated boreholes can display high mean pressures, small amplitude diurnal variations, and mean pressure trends dissimilar to those shown by most of the correlated sensors within the cluster. Such correlated boreholes resemble anti-correlated ones in every aspect but their phase. The black line in Fig. 10 shows an example of this unusual kind of correlated pressure record.

**3.2   Diffusivity at the glacier bed and the two-dimensional nature of the drainage system**

Phase lags between hydraulically connected boreholes, as well as the changes in the amplitude of diurnal variations, are the signature of the propagation of pressure waves through a diffusive system (Hubbard et al., 1995; Werder et al., 2013). Their study allows us to assess to what extent diffusion processes (with finite diffusivity) control the propagation of pressure variations in the subglacial drainage system. Phase lags for each time series within a cluster were computed relative to the mean
diurnal residual of the cluster, and the associated lag corresponds to the time offset that maximizes its correlation coefficient with the mean diurnal residual of the cluster.

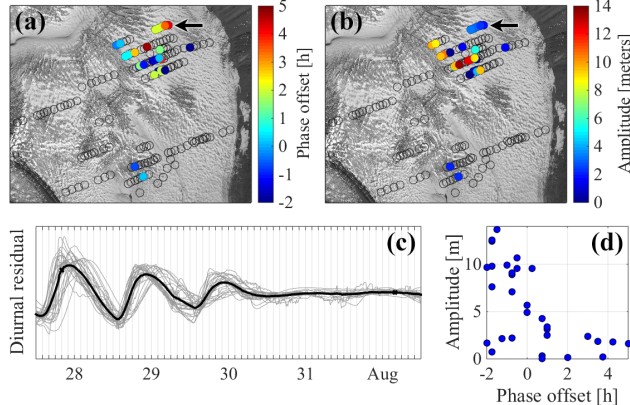

**Figure 14.** Spatial distribution of phase lag (a) and amplitude of diurnal variations (b), for the correlated boreholes in window g (July 27th to August 2nd, 2015) of cluster H1 (see Fig. 10). Panel c shows the individual diurnal residuals for each borehole as thin lines, and the mean diurnal residual for the cluster as a thick black line. Panel d shows the relationship between phase lag and amplitude.

The phase lags in mechanical clusters are often very small and close to our measurement error. However, in hydraulic clusters, we consistently observe phase lags up to six hours. Note that our clustering method suppresses the clustering of time series with time lags around 6 and 18 hours, as those would neither be well correlated nor anti-correlated. However, manual inspection of the boreholes excluded from cluster H1 and other large clusters suggests that lags larger than 6 hours are

extremely rare. Figure 14 shows the distribution of phase lags and amplitudes of diurnal variations for the correlated sensors in window g of cluster H1 (see Fig. 10). A diffusion model for pressure variations would predict that observations at increasing distances from an active drainage axis, such as a channel, would display increasing phase lags and decreasing amplitudes (Hubbard et al., 1995). In general, we indeed observe that leading phases in correlated boreholes tend to be associated with larger amplitudes. We present a typical example of this loose relationship in Fig. 14d. In this example, as well as in most cases,

sequences of boreholes that clearly display a diffusive behaviour are the exception. One example of behaviour qualitatively consistent with diffusion is the line of four boreholes pointed by a black arrow in the upper right corner of panels a and b of Fig. 14. By contrast, most groups of boreholes display a more complicated pattern of phase lag and amplitude distribution. In other exceptional cases, there are even groups of boreholes where we observe increasing phase lags accompanied by increasing amplitudes, opposite to what we would expect in a diffusive system.

For each of the many spatial patterns shown by the different clusters, we also evaluated whether they were compatible or not with a subglacial drainage system on which horizontal conduits are confined to the bed interface only. We found that in some cases, the clustered boreholes exhibit a structure seemingly incompatible with a two-dimensional drainage system. Figure 15 shows an example of five clusters where the clusters in panels c, d, and maybe b, seem to straddle the one on panel f.

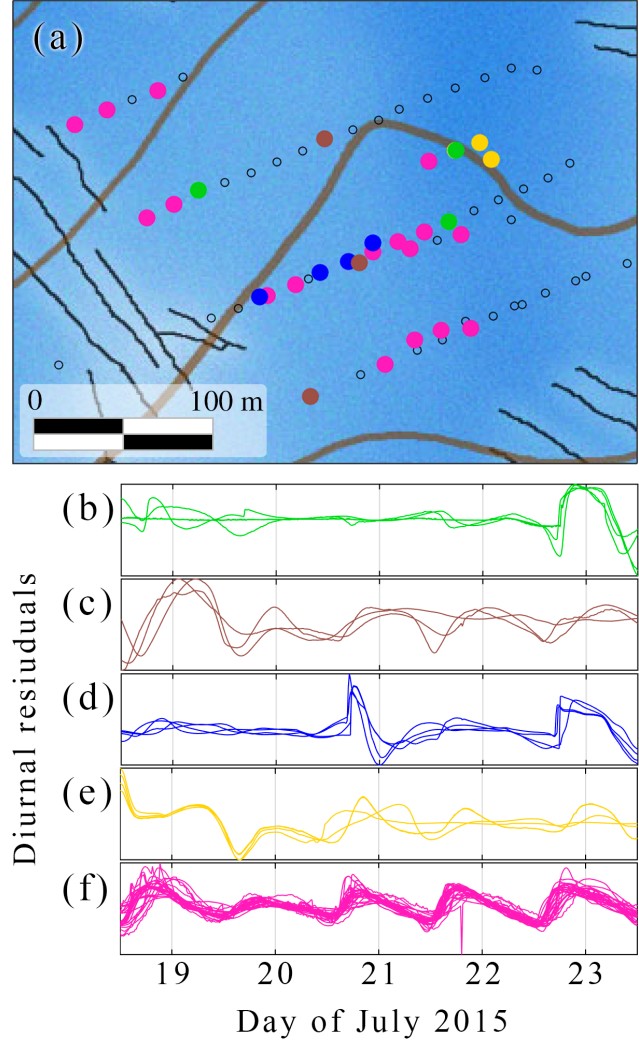

**Figure 15.** Detailed spatial distribution (a) and diurnal residuals (b-f) of five clusters observed in the plateau area between July 18th and 23rd, 2015. Only correlated boreholes are shown.

## 3.3 Spatial patterns of connected and disconnected areas

Assuming that two boreholes that are correlated members of the same hydraulic cluster are linked by a hydraulic connection, then our records show clearly that some regions of the bed are more susceptible than others to forming hydraulic connections. On the other extreme, some regions seem to remain disconnected through the multiple years we have data for. However, quan-

5  tifying these differences in connectivity requires us to account for the spatial sampling bias of our dataset adequately. For this reason, using the whole dataset we have calculated the average probability of a hydraulic connection between two boreholes. This probability was calculated under the assumption that the bed is homogeneous, meaning that hydraulic connections are





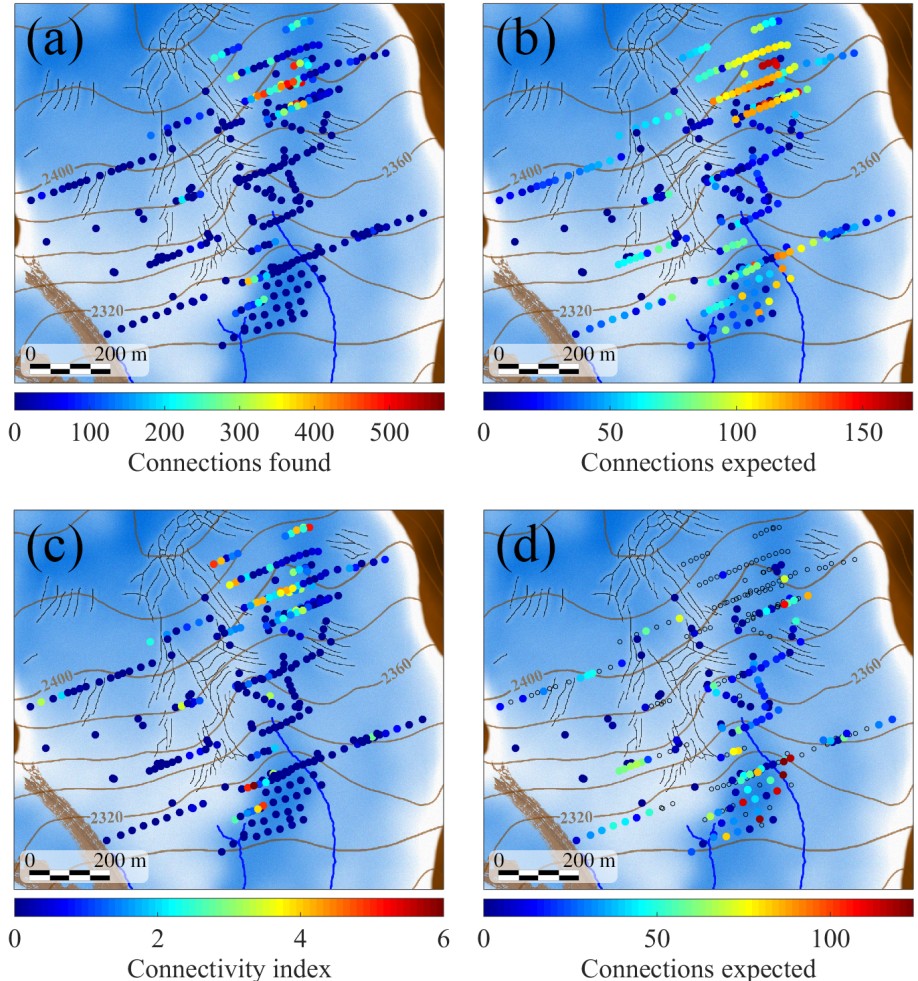

**Figure 16.** (a) Total number of hydraulic connections observed in each borehole. (b) Number of hydraulic connections expected for each borehole using the estimated connection probability $P$ (see Eq. 3). (c) Connectivity index for each borehole. (d) Number of hydraulic connections expected for borehole where no connections were observed.

equally likely anywhere along the bed. Here we contrast the predictions of the connection probability computed in Section 2.5 with the observed number of identified hydraulic connections at each position, allowing us to test how heterogeneous the drainage system is.

Figure 16a shows the total number of hydraulic connections found in all windows of our clustered dataset for each borehole.
5 However, this number is heavily biased by our spatial sampling and data availability. Figure 16b shows the number of connections that we expect for each borehole using the connection probability $P$ (see Section 2.5) and actual data availability at each borehole. We can see that there are two areas where we would expect the highest number of connections: a large one in the upper right of the study area (the plateau), and a smaller one at the bottom near the eastern surface stream. Nonetheless, only the





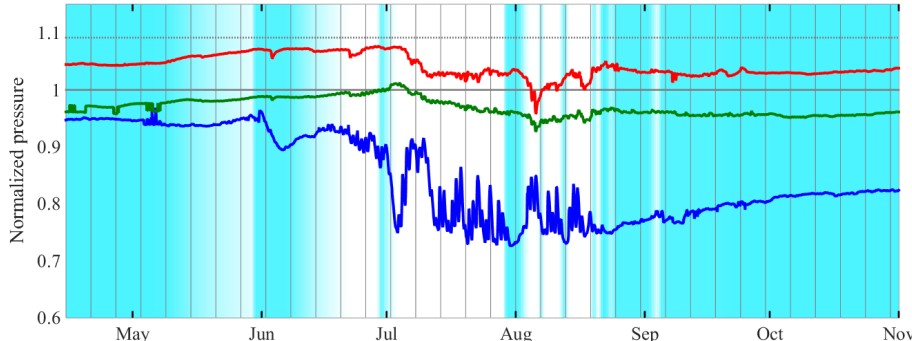

**Figure 17.** Spatially averaged mean pressure values for three types of boreholes between May 15th and November 1st, 2015. In blue, the mean of 171 boreholes that at some point in the 2015 melt season were hydraulically connected. In green, the mean of 78 boreholes that participated in mechanical clusters but were never hydraulically connected. In red, the mean of 33 boreholes that were at some point anti-correlated members of a hydraulic cluster but were never hydraulically connected. All pressure values used to compute these means were normalized by the overburden pressure of each corresponding borehole.

one at the plateau actually shows a large number of connections (Fig. 16a). To better explore this difference, we have defined a connectivity index consisting of the ratio between actual and expected connections. Figure 16c shows the connectivity index for all boreholes. We observe that some regions display 2 to 6 times more connections than expected. In particular, the region at the top of the study area, and the one at the bottom between the two surface streams.

Finally, Fig. 16d shows the number of expected connections for boreholes where we found no connections at all. We can see that some regions include multiple boreholes for which we would have expected to observe over a hundred connections but found none. The most notable example is the area next to the eastern surface stream.

### 3.4   Spatially averaged pressure trends

As we have pointed out, we cannot apply our clustering algorithm outside of the summer melt season due to the lack of diurnal
forcing. However, a general overview of seasonal pressure changes through the year, can be obtained by averaging over the records of all sensors based on their behaviour during the melt season (see s ection 2.6 for the averaging method). We have selected three types of boreholes whose means are displayed in Fig. 17:

1.  Boreholes that we identified at some point as correlated members of a hydraulic cluster (in blue). We expect these boreholes to be representative of the regions of the bed over which the summer drainage system develops during the melt
season.

2.  Boreholes that were anti-correlated members of a hydraulic cluster, without ever becoming hydraulically connected (in red). If these anti-correlated pressure variations are the result of horizontal normal stress transfers, the corresponding boreholes must be necessarily sampling disconnected portions of the bed. Therefore, they constitute our best proxy of

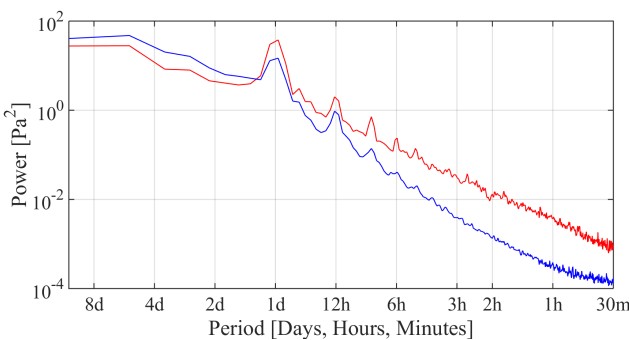

**Figure 18.** Power spectrum of clusters H1 (in blue) and M1 (in red).

the pressure variations in such disconnected areas. We have excluded other disconnected boreholes due to the concerns that some of them might be encased in ice and not directly sampling the pressure at the bed.

3. Boreholes that we identified at some point as members of mechanical clusters and were never hydraulically connected (in green). We include this category to provide more information for the interpretation of mechanical clusters.

We can see how the three types of boreholes display mostly constant pressure before and after the melt season, differing in their mean value by up to 25% of overburden. Connected boreholes (in blue) show lower pressures, with significant pressure variations during the melt season and a post-season value about 15% lower than the pre-season mean value. In contrast, mechanical and anti-correlated boreholes show very similar pre and post-season mean values. The mean pressure in mechanical and anti-correlated boreholes remain close to the overburden pressure, however the former are generally below it and the latter
above.

Pressure variations in "mechanical" and "hydraulic" cluster can also be studied in the frequency domain. We have seen that mechanical clusters are characterized by more square-wave-shaped diurnal variations, as is clear in the shapelet of Fig, 5, and these variations have small amplitudes, typically below 2 meters. Spectrally, the time series produced by mechanical clusters also have a larger high-frequency content than hydraulic clusters. To quantify the difference between cluster types, Fig. 18
presents the average power spectrum of the pressure time series in cluster M1 (in red) and H1 (in blue). For M1, the data comes from 30 boreholes providing 492 time series on 44 different 6-day time windows, and for H1 it comes from 120 boreholes providing 865 time series on 33 different 6-day time windows. In each window, all time series belonging to H1 or M1 over that window were tapered using a turkey window (Bloomfield, 2004) with $\alpha = 1/3$ and then fourier transformed. The power spectra over all windows and available time series therein for each cluster were averaged to produce the H1 and M1 average
spectra.

We can see how pressure variations in the mechanical cluster M1 have a greater power that those of H1 for all periods shorter than one day. Conversely, cluster H1 has a greater power than M1 for all periods longer than two days, reflecting a greater amplitude of low-frequency variability.





## 4 Discussion

We can robustly automate the picking of clusters based on the similarity of diurnal residuals, which correspond to the normalized residuals of the raw pressure signals relative to a diurnal running mean (Eq. 1). The algorithm is based on hierarchical clustering and an "absolute" version of the Euclidean distance metric (Eq. 2). This distance metric defines how the similarity between time series is quantified.

Different clusters differ in two respects: the details of the shape of diurnal pressure oscillations (in terms of properties such as how sharp the daily pressure peak is), and the day-to-day variations in the amplitude of the the diurnal pressure oscillations. We have chosen to normalize pressure variations (that is, not to take into account the absolute amplitude of pressure oscillations) and to group boreholes that are both well correlated and anti-correlated with each other.

We can distinguish two cluster types, which we have suggestively named "mechanical" and "hydraulic". The former is characterized by a more square-wave shape of the diurnal oscillations (Fig. 5) with a significant high-frequency content (Fig. 18) and small amplitude diurnal oscillations, typically 1–2 m (Rada and Schoof, 2018). In contrast, the hydraulic cluster type is characterized by smoother pressure variations that can reach large amplitudes (typically tens of meters). Both cluster types can also often be broken into two subclusters of mutually anti-correlated pressure records. Hydraulic clusters differ from their mechanical counterpart in having one "correlated" subcluster for which the raw pressure records have significantly higher effective pressure and larger diurnal amplitudes compared with the other, "anti-correlated" subcluster. For mechanical clusters, effective pressures and diurnal amplitudes are relatively small and comparable on both subclusters, which also differ in their phase (see Fig. 7).

We interpret hydraulic correlated subclusters as consisting mainly of boreholes that are physically connected to an active subglacial drainage system. This connection is consistent with their almost always below-overburden internal water pressure and the large amplitude of their diurnal pressure variations (Kamb, 1987; Hubbard and Nienow, 1997).

The anti-correlated subcluster must then correspond to hydraulically isolated boreholes that experience pressure oscillations due to normal load transfers from a nearby active drainage system. Within such an active drainage system, below-overburden water pressure increases the normal stress near these isolated boreholes, and this increase then causes the water pressure in the boreholes to rise to keep their volume fixed. Pressure variations within the active drainage system change the strength of this load transfer, with higher water pressures in the active system generating lower normal stress in the surrounding area and vice versa. This induces anti-correlated pressure variations in the isolated boreholes (Weertman, 1972; Murray and Clarke, 1995; Lappegard et al., 2006). Some pressure records suggest that the load-transfer mechanism can exceptionally generate correlated pressure variations (see the black line on Fig. 10). We speculate that this reflects a "second-order" load-transfer. In these cases, an active drainage system would induce anti-correlated pressure variations in an isolated water pocket, and this water pocket would, in turn, induce pressure variations on a second isolated water pocket. These later variations would then be correlated with those of the active drainage system. This kind of interaction would be possible only if the first water pocket extends far enough from the active drainage system, such that its influence on the second isolated water pocket becomes stronger than that of the active drainage system.



Using a semicircular R-channel model, Weertman (1972) showed that the load transfer effect extends over a distance similar to the radius of the channel. In such a case, the probability of randomly drilling into a channel would be the same as that of randomly drilling into a section of the bed under the influence of load transfers from the channel. However, in cluster H1 we observe that anti-correlated boreholes account typically for 20% to 40% of the cluster, numbers that are similar to what

we observe in other large clusters. While this difference could arise from anti-correlated boreholes being detectable only at distances shorter than one channel radius, the distribution of correlated and anti-correlated boreholes suggests that large clusters are not composed of a network of well developed R-channels. In Fig. 10f–h, we can see how anti-correlated boreholes tend to appear in groups surrounding the areas dominated by correlated boreholes, while for a network of R-channels we would expect them to be finely interleaved in between the channels (assuming that the diameter of those channels would be much smaller

than our 15 m sample spacing). Therefore we interpret large clusters, such as H1 in windows e to h of Fig. 10, as consisting of a distributed drainage system where the gaps between conduits are generally small compared with the borehole bottom diameter, that we estimate to be between 25 and 50 cm approximately. These conduits could correspond to a network of cavities, pore spaces, small channels, or a combination of them. A further support of this interpretation, is the clear contrast in the pressure variations within hydraulic clusters with those observed within a large channel, such as the one we observed on the 2013 melt

season and extensively described in Rada and Schoof (2018).

We interpret "mechanical" boreholes as similarly isolated boreholes that experience pressure oscillations due to changes in the stress field around them. This is consistent with their typically near-overburden water pressure (Fig. 17), their widespread spatial distribution, and their greater high-frequency content (Fig. 18). The diurnal acceleration and deceleration of the glacier could induce the changes in the stress field that drive the pressure variations within mechanical clusters (Andrews et al., 2014).

If that is the case, their characteristic "square-wave" summer profile of diurnal variations (Fig. 5), would be suggestive of stick-slip basal motion. Arguably, such stick-slip events would induce well correlated pressure variations in all cavities, making difficult to account for anti-correlated records. However, if the water pockets associated to mechanical clusters are not fixed to the bed but advected with the glacier sole, such anti-correlated pressure variations could be derived from differences in ice-bed convergence velocities. Experiments by Thompson et al. (2020) show that all basal clasts have an associated water pocket

around them. Such water pockets, as well as any other water pocket advected at the glacier sole (such as the one created by the borehole itself), would experience an increase in internal pressure if accelerated on an area of ice-bed convergence, and a decrease in pressure if accelerated on an area of ice-bed divergence.

Basal conditions surrounding boreholes belonging to mechanical clusters or hydraulic anti-correlated subclusters would be essentially the same. However, the higher mean pressure of the latter (see Fig. 17), is consistent with the increased normal

stress experienced in the proximity of an active drainage system operating at a relatively low internal water pressure (Fig. 1).

### 4.1 Diffusivity distribution at the glacier bed

Our observations suggest that hydraulic connections are more likely to be found in some areas of the bed than others (see Fig. 16). The areas with a high connectivity index cannot be predicted by simple upstream area calculations that assume that the effective pressure is a constant fraction of the ice overburden pressure, and do not display a strong association with basal





or surface topography features. We found the largest concentration of boreholes with high connectivity index on the plateau area (located in the northeast corner of the study area, see figure 2 of Rada and Schoof (2018)). This area is characterized by a relatively flat surface and low-angled bedrock topography. However, other similarly flat areas do not show enhanced connectivity. A more detailed analysis to explain these differences and the overall location of enhanced connectivity areas

would require further information about the points were meltwater supply enters the subglacial drainage system.

In contrast with the connected areas, a significant fraction of the glacier bed can remain disconnected year-round, even during the spring event. The concentration of permanently disconnected boreholes in areas that we sampled densely and over long periods (see Fig. 16d) suggests that their presence is robust and we cannot attribute it to sampling biases. Such disconnected areas could have a significant effect on the mean effective pressure at the glacier bed and thus on basal sliding rates. Mejia

(2021) suggests that the seasonal variations of surface speed in Greenland might be controlled in part by changes in the extent of disconnected areas.

The location of permanently disconnected areas does not seem to be defined by the local geometry of the glacier either. We find the two most notable cases located beside the two areas with the highest connectivity. This association could suggest that areas with high connectivity might turn nearby sections of the bed into disconnected areas: in the absence of local water input,

the highly connected parts of the bed might able to evacuate all incoming water from further up-glacier and would not require the adjacent disconnected areas to form part of the drainage system.

Hoffman et al. (2016) also recognized a marked heterogeneity in the diffusivity of the glacier bed, proposing the existence of a disconnected or weakly connected component. In Rada and Schoof (2018) (see figure 8), we identified some features in the data that might be consistent with a slow leakage from the disconnected parts of the bed. The boreholes that most confidently

represent the pressure variations in the disconnected portions of the bed are the anti-correlated boreholes of hydraulic clusters because, if our interpretation is correct, such condition guarantee that they are sampling the bed-ice interface. Interestingly, the mean pressures in such boreholes (see red line in Fig. 17) does not show a significant leakage during the period in which low pressures dominate the connected drainage system. In a water volume that is hydraulically disconnected except for a slow leakage, the associated reduction in water volume should show up as a gradual drop in water pressure. Alternatively, if the

water pressure remains constant, the leakage should result in a reduction of the water volume within these disconnected areas of the bed. However, we do not see that the pressure in the red line of Fig. 17 drops in response to low water pressures within the connected drainage system. Similarly, Fig. 11b does not show a reduction of the number of hydraulic anti-correlated boreholes in those periods of low water pressure within connected boreholes.

Additionally, the anti-correlated boreholes are in the proximity of the connected portions of the bed, arguably a factor that

should increase the leakage rate, and the lack of a slow pressure response indicating leakage suggests that disconnection is in effect complete. Alternatively, we can argue that the high effective pressure in the connected areas would favour the closure of connections in the surrounding bed due to bridging stresses (Weertman, 1972; Lappegard et al., 2006), therefore, making disconnected areas less likely to leak if they are close to hydraulically connected ones. Therefore, these observations suggest that there is either no significant leakage, or that ice creep is capable to keep high pressures despite of the leakage.





## 4.2  Subglacial drainage evolution

The extent of the disconnected fraction of the bed changes through the melt season. For the 2015 season, Fig. 11 suggests that the connected fraction of the bed increases quickly at the start of the melt season in response to the initial rise in meltwater supply in the last week of May. However, the lack of a variable meltwater supply before that period would have rendered any

preexisting connection undetectable by our method. Therefore, we do not know if the increase in observed connections is due to the establishment of new connections or an increase in the ability of our method to detect them.

We can remove this uncertainty by observing the evolution of the subglacial drainage during long periods of sustained diurnal meltwater supply. There were two of such periods in 2015: the second half of June, and most of July (see Fig. 11 panel c). Notably, we observe a different behaviour in each one: during the second half of June, the connected areas of the bed undergo

sustained growth. In contrast, during July we observe a slow decline in the extent of connected areas. These two behaviours contrast in several other important aspects. (1) In the earlier period, the drainage system starts small and fragmented, while the later one starts as a single large connected subsystem. (2) In the earlier period, the drainage system grows and preserves or reduces its degree of fragmentation, while we observe the opposite trend in the later period. (3) The earlier period is characterized by high and relatively constant mean water pressure, similar to winter pressures (see Fig. 17). In contrast, the

later period starts with a significant drop in diurnal mean water pressure, followed by a sustained downward trend (see Fig. 10). (4) The earlier period is characterized by diurnal variations of much smaller amplitude than those in the second period. (5) In the earlier period, there is not a clear trend in the fraction of anti-correlated boreholes in hydraulic clusters, while in the second period there is a clear increase of anti-correlated boreholes relative to the correlated ones.

We interpret the transition between these two behaviours, as a turning point in the efficiency of the drainage system, possibly

associated with the onset of viscous heat dissipation as the dominant term for conduit growth (Röthlisberger, 1972; Schoof, 2010). Therefore, this transition would mark the beginning of the "channelization" of the drainage system, a process that is consistent with the sustained decrease in water pressure (see Fig. 10), and the increasing fragmentation.

The increase in the fraction of anti-correlated boreholes is also consistent with the perimeter enlargement associated with the development of an arborescent drainage system. It remains unclear whether the snowfall event that separates both periods

played a significant role in triggering this transition.

The difference between these two periods suggests that sustained meltwater supply might have a different effect on the development of the summer drainage system, enlarging the area occupied by the low-efficiency drainage system found early in the season, yet promoting fragmentation and focusing in the more efficient drainage present later in the season.

The lack of variable meltwater supply outside the melt season hinders the application of our method. However, widespread

near-overburden water pressures and insignificant correlation between pressure changes when they happen, suggests that any drainage system that persists over winter is highly fragmented and mostly disconnected from the surface.

While the spatial structure of the clusters identified in most time windows can be described using a two-dimensional conduit network, some clusters seem to be topologically disconnected, such as clusters d and f of Fig. 15. Explaining the structure of these clusters would require a subglacial drainage system that includes horizontal englacial conduits at multiple levels.





However, this topological disconnection might be merely highlighting a shortcoming of our clustering technique. In particular, the method is unable to track changes in the drainage system at time scales shorter than the time window used for the clustering process. Therefore, cluster d could consist of mutually connected boreholes that connect and disconnect from cluster f as a consequence of a switching event (Rada and Schoof, 2018). In that case, cluster d would not constitute an independent drainage

subsystem, but a temporary extension of cluster f. We consider that the evidence provided by Fig. 15 and other examples of clusters seemingly incompatible with a 2-D structure of the subglacial drainage system are not strong enough to discard that interpretation. However, they are suggestive of some degree of three-dimensional structure.

## 4.3 Methodological caveats

Our clustering removes information about the mean water pressure and the amplitude of diurnal oscillation through the pre-

processing step of forming normalized diurnal residuals. This step is necessary to identify mechanical clusters, and to incorporate all anti-correlated holes of a hydraulic cluster. Nonetheless, absolute pressure variations are relevant to the question of whether two boreholes have an actual hydraulic connection, which is one of the main objectives of our study.

While differences in absolute pressure can arise from differences in the absolute elevation of the lower end of the boreholes or sensor calibration errors (see section 1 of the supplementary material of Rada and Schoof (2018)), two boreholes with

well-matched diurnal residuals but with different oscillation amplitudes necessarily experience variations in hydraulic head differences that themselves resemble the diurnal residuals. For two hydraulically connected boreholes, such a hydraulic head difference implies that water will flow. If also there is water storage along the flow path with high water pressure corresponding to larger storage (Freeze and Cherry, 1979; Hubbard and Nienow, 1997; Werder et al., 2013) then we expect to see an attenuation in oscillation amplitude and an increasing phase lag along the flow path.

However, our clusters do not always conform to this expectation (see Fig. 14). While in general phase-leading boreholes display larger amplitude of diurnal variations, suggesting that diffusion processes do play an important role in the propagation of pressure signals. There are numerous cases where phase lags and the amplitude of diurnal variations do not follow the pattern expected in a diffusive system. Those cases suggest that the spatial resolution our data is unable to distinguish the heterogeneities of the distribution of basal diffusivity. This shortcoming suggests that the diffusivity distribution has a fine

structure at scales smaller than the minimum spacing between our boreholes ( 15m). Large tortuosities and abundant englacial connections could also contribute to the complex patterns of phase lags and amplitudes we observe.

In a system dominated by diffusive pressure signals, our clustering technique would also be a poor choice due to the significant phase lags that can be introduced by diffusion. However, we also tested other clustering variants that should perform better in that scenario, yet they proved to do a worse job in reproducing our manually picked clusters (see supplementary

information). For example, the running standard deviation pre-processing quantifies variations in amplitude but is insensitive to phase lags. Similarly, the DTW distance metric assign small distances to signals with similar shapes, regardless of phase lags or stretching. The lack of diffusive signals is also consistent with what we observed during the manual picking of clusters for the calibration, validation and testing datasets.





The stark contrast between the small number of apparently diffusive signals observed at South Glacier and those predicted by models is unlikely to arise from the lack of diffusion processes at the bed. Instead, it is most likely a result of the simple conduit geometries assumed by models, such as sheets or straight lines between grid nodes. Therefore, this discrepancy also points to a diffusivity distribution that has a fine structure that we cannot resolve with a 15 m sample spacing.

Another shortcoming of our clustering technique is the disregard of diurnally-averaged pressures. While these are frequently uncorrelated for mechanical clusters and anti-correlated subclusters, our method can also spuriously identify boreholes with poorly correlated diurnally averaged pressure as hydraulically connected, so long as their diurnal residuals resemble each other closely enough. Theoretically, the diffusive picture of the drainage system would suggest that at medium-term timescales (on which the conduit configuration and diffusivity do not change) pressure variations should correlate. Our clustering can thus

produce false positives for hydraulic connections, as the black line on Fig. 10, that might instead be the result of a second-order load transfer.

We should recall that our method relies on the ability of each drainage subsystems to modulate the forcing signal distinctly, as a result of their specific geometry, permeability, and storage distribution. The high fragmentation of the subglacial drainage observed during some periods suggests that indeed a different subsystem often produce a significant and distinct modulation

of the forcing. However, such apparent fragmentation could also arise from differences in the forcing itself.

We have observed this phenomenon on rare occasions while manually identifying clusters. In such cases, we have found boreholes that display similar pressure variations but are very far apart across the glacier, a geometry that makes a hydraulic connection improbable, especially if there are no other boreholes in-between showing similar pressure variations. These cases of similarity that is likely the result of common forcing are expected to be more frequent between nearby boreholes, a situation

in which we would be unable to distinguish this phenomenon from true hydraulic connections. Therefore, we expect that some of the identified connections are artifacts due to the similarity of the forcing signal in individual drainage subsystems.

## 5   Conclusions

We were able to automatically pick clusters of boreholes based on the similarities between their pressure response to surface meltwater supply, and we classified these clusters into two main types: hydraulic and mechanical. Both clusters types are often

composed of two subclusters of mutually anti-correlated boreholes. For mechanical clusters, the two subclusters differ only in their phase, while in hydraulic clusters one subcluster shows higher mean water pressure and diurnal oscillations of smaller amplitude. We refer to this subcluster as anti-correlated because it displays pressure variations that are anti-correlated with the surface meltwater supply.

We interpret correlated boreholes of hydraulic clusters as being hydraulically connected to the surface meltwater supply,

while anti-correlated boreholes sample disconnected areas of the bed. These disconnected areas can display small water pressure variations due to normal stress transfers associated with the pressure variations within correlated boreholes (Weertman, 1972; Murray and Clarke, 1995; Lappegard et al., 2006). In large hydraulic clusters, we generally find anti-correlated boreholes at the edge of groups of correlated boreholes, suggesting that the distributed drainage system associated to these clusters





is composed of a network of small conduits with spacings smaller than the borehole bottom diameter (approximately 25–50 cm). Within these hydraulically connected areas of the bed, patterns of phase lag and amplitude attenuation suggest that the diffusivity distribution at the bed presents a fine structure at scales smaller than our minimum borehole spacing of 15 m.

Boreholes in mechanical clusters are also disconnected from the surface meltwater supply, and their pressure variations are likely to be controlled by stress changes associated with the glacier motion. In this case, the square-wave shape could be suggestive of a stick-slip motion regime. Anti-correlated signals in mechanical clusters also suggest that some of boreholes sample water pockets that move with the glacier, attached to the glacier sole instead of the bedrock.

The distribution of areas of the bed connected to or disconnected from the surface meltwater supply changes throughout the year, and even during the melt season. Some areas of the bed can show a large number of hydraulic connections while others remain disconnected year-round. The distribution of these areas does not seem to be dictated by surface and bed topography alone. We hypothesize that the location of the meltwater supply input points plays an important role in determining which parts of the bed are well-connected or disconnected. Disconnected areas do not show a significant water leakage during the melt season, suggesting that the hydraulic disconnection is complete. However, if bridging stresses are a significant contributor to hydraulic disconnection around connected conduits (Weertman, 1972; Lappegard et al., 2006), it is possible that leakage can become significant in disconnected areas unaffected by normal stress transfers.

The evolution of cluster sizes and fragmentation of the drainage system during the melt season suggest that repeated diurnal pulses of meltwater supply promote the growth of the low-efficiency drainage systems found early in the season while stimulating the shrinkage, fragmentation, and focusing of the more efficient drainage systems that appear later in the season. Therefore, the increase in drainage efficiency would inhibit the growth of the connected areas of the bed. In 2015 at South Glacier, the transition between these two regimes took place during the first days of July, when the pressure within connected boreholes underwent a significant pressure drop (see Fig. 10). This turning point might be associated with the onset of viscous heat dissipation as the dominant term for conduit growth. It is important to note that the more efficient drainage systems we refer here, do not correspond to a fully channelized drainage system as the one we observed in the summer of 2013 and described in Rada and Schoof (2018). Instead, they correspond to a somewhat more efficient or more channelized distributed drainage.

Our observations support some of the features shown by recent subglacial drainage models (Schoof, 2010; Hewitt, 2011; Schoof et al., 2012; Hewitt et al., 2012; Hewitt, 2013; Werder et al., 2013; Bueler and van Pelt, 2015), such as the existence of a distributed drainage system early in the melt season that gradually evolves into a progressively more channelized and focused system. However, the most notable difference with the models is the extremely heterogeneous distribution of diffusivity that our results suggest, and the robust support for the existence of disconnected areas. These disconnected areas invalidate the assumption of these models that the distributed drainage system pervades the whole glacier bed. Therefore, in addition to the effective pressure within the connected parts of the drainage system, the extent of this system could also be an essential control on basal speed variations. It is possible that even relatively small disconnected areas could have a disproportionate effect on basal speed. Also, while our observations cannot confirm or refute the year-round persistence of a distributed drainage system, winter pressure variations suggest high fragmentation of the drainage system, imposing a limit to the extent of such a persistent distributed drainage.



*Data availability.* The presented dataset will be made publicly available in the future. Ongoing work is taking place to meet the format and create the ancillary data and documentation required for the release, that is expected to happen fully or partially by 2022. In the meantime, it is available on request from the second author at cschoof@eoas.ubc.ca.

*Competing interests.* The authors declare that they have no competing interests

*Acknowledgements.* We want give special recognition and thanks to Lance Goodwin, who was a big enthusiast of our scientific work at South Glacier and was always ready to help with a big smile. Lance have left us way too soon, and we deeply feel his departure. We thank also Manar Al Asad, Faron Anslow, Ashley Bellas, Kyla Burrill, Emilie Delaroche, Jennifer Fohring, Tom-Pierre Frappé-Sénéclauze, Johan Gilchrist, Marianne Haseloff, Ian Hewitt, Marc Jaffrey, Alex Jarosch, Conrad Koziol, Natalia Martinez, Arran Whiteford and Kevin Yeo for assistance in the field. Gwenn Flowers provided bed elevation, South Glacier AWS data as well as continuous help and advice without

which this project would not have succeeded. Additional AWS data were made available by Christian Zdanowicz and Luke Copland. We are indebted to Parks Canada and Kluane First Nation for their support and permission to operate at the field site, to Doug Makkonen, Dion Parker and Ian Pitchforth for expert flying, to Andy Williams, and Sian Williams for logistics support. This work was supported by the Natural Science and Engineering Research Council of Canada through Discovery Grants 357193-08 and 357193-13, Accelerator Supplement 446042-13, Northern Research Supplements 361960-06 and 361960-13 as well as Research Tools and Instruments Grant 376058-09; by the

Polar Continental Shelf Project through grants 625-11, 638-12, 637-13, 663-14 and 667-15; and by the Canada Foundation for Innovation and British Columbia Knowledge Development Fund through Leaders Opportunity Fund project number 203786 and 227698.



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
