# Peer review of "Channelised, distributed, and disconnected: Spatial structure and temporal evolution of the subglacial drainage under a valley glacier in the Yukon"

_The Cryosphere, 2022_

## Referee Comment (RC1)

**Review of the manuscript entitled "Channelised, distributed, and disconnected: Spatial structure and temporal evolution of the subglacial drainage under a valley glacier in the Yukon", by Rada and Schoof.**

**1 General comments**

This study by Rada and Schoof describes a new method to study the evolution of the subglacial drainage system of a small mountain glacier through the analysis of a large number of pressure records. The authors first present in great details the processing framework that is used to identify cluster of related pressure records and the evolution of these clusters through time. They then present the results of their analysis for the year 2015 when the pressure record was covering the largest part of the glacier for an extensive period of time.

I found the description of the processing framework to be very clear and well written, each step was carefully described and it would be easy from this work to implement this process to a new dataset. The authors made a good use of the supplementary material to present the alternative methods that were discarded which gives a good overview of the superiority of the methods that are presented in the main manuscript. After this great presentation of the methods, I was a bit disappointed by the results and discussion part of the manuscript. I however think that this is more due to the structure of the result and discussion section rather than from the content of those parts of the paper. I find that the conclusion of the study are not as clear as they should be and that some reformatting could help with that. In the Result part, the structure of the different subsection does not seem to be the most logical to me, for example, the spatially averaged pressure trends are discussed both in Figure 10 and 17 in two different section, I feel that the first sub-section on the evolution of the subglacial drainage system should focus either on the evolution of the pressure records or the spatial distribution of the subglacial drainage system. As it is know the mix between both spatial and pressure evolution makes the results harder to read in my opinion. Some grouping of results also do not seem completely logical to me, for example in section 3.2 I would rather see only the analysis on diffusion while it

seems to me that the two-dimensional nature of the drainage system would fit better in section 3.3. The Discussion part presents a lot of interesting points, and I feel that a short conclusion with the major takeaways at the end of each section will help to point out the main findings of the study and clarify the results of the study.

Bellow are a few more suggestion and questions on the manuscript in general

1. Regarding the length of the time window, I wonder if using several different time windows with different length would yield more information when comparing their results in term of clustering?

2. I feel that the discussion between correlated and anti-correlated series should be made clearer earlier in the manuscript. The process itself is well illustrated in Figure 1, but I feel that the author are missing an opportunity to clarify their workflow when they introduce the equation for the absolute Euclidian distance where the reason for the use of this specific formulation could be reiterated.

3. At some point in the manuscript, I was not sure if Pressure was designating water pressure or effective pressure, which is a major issue when describing increase or lowering of the pressure. I urge the authors to use either effective pressure or water pressure throughout the manuscript which would help with readability.

4. On the spatial distribution of the disconnected regions, I was wondering if they were appearing consistently in the same region for the different years, and if that is the case, are there any velocity records that they can be compared against?

**2    Specific comments**

Bellow is a list of more specific and technical comments throughout the manuscript given with line (L) and page (P) numbers:

- L14-P1: "diffusivity" has a typo.

- L7-P2: The references here all refer to ice-sheets velocity, given the fact that the present study treats of a mountain glacier, references pertaining to this type of glaciers might be better suited.

- L17-P2: "OBP" is defined here but used only once in the text, perhaps it should be omitted and only described in the caption of Figure 1.

- L28-P2: The citation of models here is strange, perhaps adding an "e.g" with a shorter list, or a review paper such as de Fleurian et al. (2018); Flowers (2015) would be better suited here.

- L18-P3: "water pressure" should be stated here, or effective pressure (see comment 3 above).

- Fig 1: Colourblind readers might struggle with the colorscheme of the arrows, perhaps something more contrasted would fit better (gradient of blue to red with black for overburden). In the caption of the figure OBP should be described.

- L8-P5: It should be "not" not "nor".

- L27-P5: the recent paper from Doyle et al. (2021) could be cited here too.

- Equation 2: There is an extraneous right parenthesis.

- L8-P13: It would be nice to have a quick description of the shapes of the pressure record for each cluster here.

- L13-P14: The colour coding for correlated and anti-correlated subclusters could be re-iterated here.

- Equation 4: Subscript $_i$ is used both for time and the number of valid sample $M_i$ which should be fixed.

- Figure 10: I think that clarifying between effective or water pressure is needed in the labels here and in other figures.

- Figure 10: I expect that the light blue shading is darker when there is snow cover but that should be clarified

- L4-P19: It should be specified that "the formation of a well developed subglacial drainage system, something that does not occur every year" on this specific site.

- L5-P20: I have a hard time identifying individual borehole records on Figure 10, perhaps splitting panel a with correlated and anti-correlated borehole in a different panel would help?

- L6-P20: It should be "a" not "an".

- L8-P20: The sentence starting on this line is hard to read and should be rephrased.

- L15-P20: "through time".

- L33-P20: Perhaps "in the study area" should be added here.

- L15-P23: I add to look for the meaning of "straddle" perhaps "intersect" would be better, or am I missing some of the subtleties of the wording?

- L6-P24: There could be a reference to the section where the probability were introduced here.

- L11-P26: Typo in "section".

- L15-P30: "might be able", "be" is missing.

- L24-P30: I am not sure why the discussion on creep that is made bellow is not stated here.

- L11-P32: It should be "boreholes".

- L16-P32: The sentence starting on this line is unclear and should be rephrased.

- L23-P32: "resolution of our data", "of" is missing.

- L31-P32: Shouldn't it be "assigns".

- Sup-L33-P2: "reproduce" in place of "reproducing".

- Sup-L34-P2: RIG should be defined here.

- Sup-L7-P3: EOF should be defined here.

- Sup-L8-P3: SOMs should be defined here.

- Equation S7 to 9: Shouldn't it be $a_i$ in place of $a_1$?

- Sup-L6-P14: "mechanical".

**References**

de Fleurian, B., Werder, M. A., Beyer, S., Brinkerhoff, D. J., Delaney, I., Dow, C. F., Downs, J., Gagliardini, O., Hoffman, M. J., Hooke, R. L., Seguinot, J., and Sommers, A. N. (2018). SHMIP The subglacial hydrology model intercomparison Project. *J. Glaciol.*, page 1–20.

Doyle, S. H., Hubbard, B., Christoffersen, P., Law, R., Hewitt, D. R., Neufeld, J. A., Schoonman, C. M., Chudley, T. R., and Bougamont, M. (2021). Water flow through sediments and at the ice-sediment interface beneath sermeq kujalleq (store glacier), greenland. *Journal of Glaciology*, page 1–20.

Flowers, G. E. (2015). Modelling water flow under glaciers and ice sheets. *Proc. R. Soc. A*, 471(2176):1–41.

---

## Referee Comment (RC2)

Review **tc-2022-90** *"Channelised, distributed, and disconnected: Spatial structure and temporal evolution of the subglacial drainage under a valley glacier in the Yukon"* by Rada Giacaman and Schoof

This paper proposes a comprehensive analysis of a large number of pressure measurements performed from 311 boreholes drilled between 2008 to 2015 on a Yukon glacier. This constitute a unique dataset with up to 157 simultaneously working sensors. This paper presents in detail the methodology adopted to compare and group these sensors as a function of their response to surface input melt, with the aim of drawing some pictures of the basal hydrology network. These results support the traditional view of a distribute system early in the melt season that transforms into a more channelized one, but suggesting also an highly heterogeneous distribution of the basal diffusivity as well as the existence of disconnected areas. This paper is overall very well written and easy to follow, the figure are clear and to the point and the methods very rigorous. I would suggest the authors to emphasize a bit more the uniqueness of the dataset they have acquired alors these years of repeated field measurements. If I am correct, this is the third paper based on this dataset, and I still think that few more papers could certainly comme out, continuing exploring its different aspects? This could be emphasized at the end of the present paper?

I have only minor remarks that are listed below.

**Minor remarks:**

- page 3, line 18: about the winter measurements, if most boreholes display pressure near overburden pressure for several months, one should expect high or even increasing surface velocity? Do you have observation of surface velocity all around the year that show that? May be this link with the surface velocity measurements (as mentioned in 2.1) even if I understand you don't have surface velocity in 2015?

- page 3, line 19: which size are expected to be this "water pockets"? The wording make me think to a feature that as similar vertical and horizontal dimensions, whereas I expect more a flat feature? May be "water patches" would be more appropriate here?

- page 3, line 20: sometime you are using upper case after a colon, sometime not. Here I would said that a starting a new sentence would work better as the second point is at the beginning of a new sentence (even a new paragraph).

- Figure 1: cases b, c, d are not steady, in comparison to a that can be steady. May be it should be mentioned in the caption. Also, how long do we expect these unsteady situations to last? Give order of magnitude.

- page 4, line 3: missing a verb in the last part of the sentence?

- page 4, line 12: it is obvious, but may be you should mention here that you are measuring water pressure in these boreholes?

- page 11, line 7: how sensitive are your results to this choice of a 6-day time window?

- Figure 7, caption: for c, you should mention that f is defined in Figs. 10 and 11.

- page 16, line 11: Is that that in winter the pressure is not showing any daily variations and more a monotonic signal? Any suggestion how the connection between the different boreholes could anyway be qualified?

- page 18, line 20: in this chapter?

- page 20, line 6: a small increase

- page 22, lines 12-14: don't really get the point of this isolated paragraph with the surrounding ones.

- page 26, line 11: section 2.6

- page 27, line 12: Fig. 5,

- page 32, line 23: resolution of our data?

- page 34, line 35: May be you could conclude by a bit of prospective? Which new measurements could allow a better understanding and complement these measurements? Array of seismometers? What more could still be inferred from this existing dataset?

---

## Author Comment (AC1)

Answers to comments in The Cryosphere Discussion of preprint

**Channelised, distributed, and disconnected: Spatial structure and temporal evolution of the subglacial drainage under a valley glacier in the Yukon**

Camilo Rada G. and Christian Schoof

August 20, 2022

**1 Answers to referee N°1**

**1.1 General comments**

We appreciate your positive comments about the description of the methodology, as well as your constructive feedback regarding the shortcoming of the paper structure and presentation of the results. We will restructure the results section as suggested and add a paragraph highlighting the takeaways of each of the three subsections of the discussion. We will answer your numbered list of general suggestions one by one:

1. Regarding the length of the time window, I wonder if using several different time windows with different length would yield more information when comparing their results in term of clustering?

   **R.**
   It would indeed yield more information, but it would make it more challenging to interpret due to the volume of data and its reliability. If we make the time window smaller, we would have a better temporal resolution to capture the rapid structural changes of the drainage system during some periods. However, at the same time, we would reduce the reliability of the detected connections. In the extreme of a one-day time window, our methodology would be useless, and a significantly different approach would be needed. In such a case, our method would assign almost any borehole with diurnal variations to a single cluster. Note also that our methodology doesn't do well when a borehole switch behaviour in the middle of a time window, sometimes failing to assign the borehole to either the initial or the final cluster. Such problems are more common with longer time windows, which, together with the loss of temporal resolution, are the main drawbacks of increasing the window size. Nevertheless, we tried both approaches (12 and 3 days time windows), which yielded additional useful information. However, the benefits of this information for the main points of our analysis were small relative to the increased complexity of the data presentation and discussion. Therefore, we decided to leave that information out of the discussion. We will add a paragraph mentioning that we decided to leave the information derived from longer and shorter time windows out of the discussion. Noting also that such information increased our confidence in interpreting the six-day time window that we found to strike the right balance between reliability and temporal resolution.

2. I feel that the discussion between correlated and anti-correlated series should be made clearer earlier in the manuscript. The process itself is well illustrated in Figure 1, but I feel that the author are missing an opportunity to clarify their workflow when they introduce the equation for the absolute Euclidian distance where the reason for the use of this specific formulation could be reiterated.

   **R.**
   We agree with the observation. We will introduce figure 7 on page 11 line 19, right before presenting the absolute Euclidean distance, to clarify the rationale behind this distance choice.

3. At some point in the manuscript, I was not sure if Pressure was designating water pressure or effective pressure, which is a major issue when describing increase or lowering of the pressure. I urge the authors to use either effective pressure or water pressure throughout the manuscript which would help with readability.

   **R.**
   We apologize for introducing that ambiguity, and we will revise the whole manuscript to use either "water pressure" or "effective pressure" to avoid any confusion.

4. On the spatial distribution of the disconnected regions, I was wondering if they were appearing consistently in the same region for the different years, and if that is the case, are there any velocity records that they can be compared against?

   **R.**
   Yes, permanently disconnected regions seem to persist through the years. We also have surface velocity records that will be analyzed in a follow-up paper. However, the small magnitude of the velocity differences between different areas of the glacier prevented us from directly studying the impact of disconnected regions on surface speed. Nevertheless, the impact could be measured on the stress field. An approach we have not yet attempted. We will make clear that disconnected regions persist through the years with observations. We will also mention that we will discuss their impact on surface speed in a follow-up paper.

**1.2 Specific comments**

- L14-P1: "diffusivity" has a typo.

  **R.**
  Will be fixed

- L7-P2: The references here all refer to ice-sheets velocity, given the fact that the present study treats of a mountain glacier, references pertaining to this type of glaciers might be better suited.

  **R.**
  We will include a reference to a well-studied mountain glacier.

- L17-P2: "OBP" is defined here but used only once in the text, perhaps it should be omitted and only described in the caption of Figure 1.

  **R.**
  We will remove the OBP acronym and use it only on Figure 1 as suggested,

- L28-P2: The citation of models here is strange, perhaps adding an "e.g" with a shorter list, or a review paper such as de Fleurian et al. (2018); Flowers (2015) would be better suited here.

  **R.**
  We will add e.g. and also Fleurian et al. (2018) in a relevant location in the manuscript.

- L18-P3: "water pressure" should be stated here, or effective pressure (see comment 3 above).

  **R.**
  We will change to "water pressure"

- Fig 1: Colourblind readers might struggle with the colorscheme of the arrows, perhaps something more contrasted would fit better (gradient of blue to red with black for overburden). In the caption of the figure OBP should be described.

  **R.**
  We will change the colour scheme to a colorblind-friendly choice and describe the acronym OBP here (as we will not use it elsewhere).

- L8-P5: It should be "not" not "nor".

  **R.**
  We will change it to "not"

- L27-P5: the recent paper from Doyle et al. (2021) could be cited here too.

  **R.**

  We will cite it as suggested.

- Equation 2: There is an extraneous right parenthesis.

  **R.**

  We will remove the misplaced parenthesis

- L8-P13: It would be nice to have a quick description of the shapes of the pressure record for each cluster here.

  **R.**

  We will add a sentence pointing to their jaggedness and resemblance to a square signal with a peak position just before dropping to base levels.

- L13-P14: The colour coding for correlated and anti-correlated subclusters could be re-iterated here.

  **R.**

  We will add the colour coding of each cluster type to that sentence.

- Equation 4: Subscript $i$ is used both for time and the number of valid sample $M_i$ which should be fixed.

  **R.**

  The number of valid samples is also a function of time, therefore, the subscript $i$ is correct in both cases. $M_i$ corresponds, in other words, to the number of boreholes with valid data for time i. For example, if we are averaging ten boreholes, in the first time step, maybe only 6 of them have data ($M_1$=6), then at time step 7, two additional boreholes might have data, then $M_7$=8. At the final time step, maybe most boreholes have ceased to produce data, and only three are still working, then $M_N$=3. We will rephrase the paragraph preceding Equation 4 to make this clear.

- Figure 10: I think that clarifying between effective or water pressure is needed in the labels here and in other figures.

  **R.**

  While in most cases it was specified in the caption, we agree that adding that information to axes labels would be beneficial. We will change figure 10 Y axis label to "Normalized water pressure", Figure 13 to "Water pressure", and Figure 17 to "Normalized water pressure". In this last figure, we will change also "averaged mean pressure" by "averaged mean water pressure" in the caption.

- Figure 10: I expect that the light blue shading is darker when there is snow cover but that should be clarified

  **R.**

  Your interpretation is correct; we will make it explicit in the caption.

- L4-P19: It should be specified that "the formation of a well developed subglacial drainage system, something that does not occur every year" on this specific site.

  **R.**

  Good point. We will change the wording to note that this is a feature particular to South Glacier.

- L5-P20: I have a hard time identifying individual borehole records on Figure 10, perhaps splitting panel a with correlated and anti-correlated borehole in a different panel would help?

  **R.**

  In this figure, we wanted to show the overall pattern of borehole behaviours by type more than providing a good display of individual borehole records. However, this concern is valid, and we will look for a way to make individual records easier to identify. Either by splitting into two panels as suggested or increasing the contrast between lines.

- L6-P20: It should be "a" not "an".

  **R.**

  We will change "an small increase" by "a small increase".

- L8-P20: The sentence starting on this line is hard to read and should be rephrased.

  **R.**

  We will rephrase to: "Such pressure drop would reduce the total normal stress supported by connected areas. Therefore, this unsupported load is transferred to the surrounding unconnected areas where the anti-correlated boreholes are located."

- L15-P20: "through time".

  **R.**

  We will change "trough time" by "through time".

- L33-P20: Perhaps "in the study area" should be added here.

  **R.**

  We will change "that incorporated all the connected sections of the bed." by "that incorporated all the connected sections of the bed under the study area."

- L15-P23: I add to look for the meaning of "straddle" perhaps "intersect" would be better, or am I missing some of the subtleties of the wording?

  **R.**

  We will change "seem to straddle the one on panel f", by "seem to intersect the one on panel f. In a two-dimensional drainage system, such a condition would imply a hydraulic connection between these intersecting clusters. However, the differences in their pressure records suggest no hydraulic connection."

- L6-P24: There could be a reference to the section where the probability were introduced here.

  **R.**

  We will add a cross-reference to section 2.5 (Spatial patterns in basal hydraulic connectivity).

- L11-P26: Typo in "section".

  **R.**

  We will remove the misplaced space within the word "section"

- L15-P30: "might be able", "be" is missing.

  **R.**

  We will add the missing "be".

- L24-P30: I am not sure why the discussion on creep that is made below is not stated here.

  **R.**

  We will rephrase that the mentioned reductions in volume would be associated with ice creep.

- L11-P32: It should be "boreholes".

  **R.**

  We will replace "holes" by "boreholes".

- L16-P32: The sentence starting on this line is unclear and should be rephrased.

  **R.**

  We will rephase as: "Such a hydraulic head difference implies that water will flow between two hydraulically connected boreholes. In such a case, we would expect differences in the hydraulic head when significant water storage exists along the flow path. These differences would take the form of oscillations with attenuated amplitude and phase lag".

- L23-P32: "resolution of our data", "of " is missing.

  **R.**

  We will add the missing "of"

- L31-P32: Shouldn't it be "assigns".

  **R.**

  Indeed, we will change it to "assigns".

- Sup-L33-P2: "reproduce" in place of "reproducing".

  **R.**

  We will change "that does best reproducing" by "that best reproduce".

- Sup-L34-P2: RIG should be defined here.

  **R.**

  We will replace RIG by "Relative Information Gain (RIG)" followed by the reference to the section that describe this concept in depth.

- Sup-L7-P3: EOF should be defined here.

  **R.**

  We will replace EOF by "Empirical Orthogonal Functions (EOF)," followed by the reference.

- Sup-L8-P3: SOMs should be defined here.

  **R.**

  We will replace SOMs by "Self-Organizing Maps (SOMs)," followed by the reference.

- Equation S7 to 9: Shouldn't it be $a_i$ in place of $a_1$

  **R.**

  Yes, indeed. We will replace all $a_1$ and $b_1$ by $a_i$ and $b_i$ respectively.

- Sup-L6-P14: "mechanical".

  **R.**

  We will replace the typo "machanical" by "mechanical"

---

## Author Comment (AC2)

Answers to comments in The Cryosphere Discussion of preprint

**Channelised, distributed, and disconnected: Spatial structure and temporal evolution of the subglacial drainage under a valley glacier in the Yukon**

Camilo Rada G. and Christian Schoof

August 20, 2022

**1 Answers to referee N°2**

**1.1 General comments**

We appreciate your positive comments about our manuscript and the uniqueness of the presented dataset. We were intentionally cautious of not explicitly saying that the dataset is unique and exceptional. However, your comments encourage us to put some of that caution aside. Therefore, we will make some changes to better communicate the uniqueness of the presented dataset to the reader. We will also emphasize the previous publications discussing this dataset and mention that a few follow-up papers are coming relatively soon. In particular, one that explores the effects of the subglacial drainage evolution on surface speed variations.

**1.2 Minor remarks**

- page 3, line 18: about the winter measurements, if most boreholes display pressure near overburden pressure for several months, one should expect high or even increasing surface velocity? Do you have observation of surface velocity all around the year that show that? May be this link with the surface velocity measurements (as mentioned in 2.1) even if I understand you don't have surface velocity in 2015?

  **R.**
  We do have continuous surface velocity records for the whole studied period, including 2015. However, due to the extent of the manuscript we have intentionally left out the analysis of velocity data in the context of the described temporal evolution of the subglacial drainage system. That will be the main focus of a follow-up paper. We will make the rationale explicit in the text and mention that the above paper is in preparation.

- page 3, line 19: which size are expected to be this "water pockets"? The wording make me think to a feature that as similar vertical and horizontal dimensions, whereas I expect more a flat feature? May be "water patches" would be more appropriate here?

  **R.**
  We have not explored the question of the shapes of these water pockets. However, we also guess they will have a "flat" aspect ratio. Nevertheless, given that we do not have observations or models to constrain their shape, we want to describe them as a generic water volume embedded into the ice. We will look for better terms than "pocket", but if we keep the terminology, we will explicitly mention in this section that we do not have constraints on the exact shape and size of these water pockets.

- page 3, line 20: sometime you are using upper case after a colon, sometime not. Here I would said that a starting a new sentence would work better as the second point is at the beginning of a new sentence (even a new paragraph).

  **R.**
  We will check the whole manuscript to make sure there is no capitalization after colons, except for proper nouns and acronyms. In this case, we will start a new sentence and remove the paragraph break before describing the second process.

- Figure 1: cases b, c, d are not steady, in comparison to a that can be steady. May be it should be mentioned in the caption. Also, how long do we expect these unsteady situations to last? Give order of magnitude.

  **R.**
  Cases b and c could be steady at some timescales if overall heat dissipation within the channel equals ice creep. We will refrain to give estimations of the timescales these configurations could last over, as they are highly dependent on the magnitude of the overburden pressure and other factors. Even d could be stable over long time scales under very shallow ice where ice rheology is elastic. For example, partially filled ice channels can last years at the edge of the glaciers, where streams go in or out of them. However, we will add that a is stable and d is generally unstable.

- page 4, line 3: missing a verb in the last part of the sentence?

  **R.**
  We will review the wording.

- page 4, line 12: it is obvious, but may be you should mention here that you are measuring water pressure in these boreholes?

  **R.**
  Good point. We will add a mention to the fact that what we are actually measuring in the boreholes is water pressure.

- page 11, line 7: how sensitive are your results to this choice of a 6-day time window?

  **R.**
  Following on the responses to referee N°1, we will elaborate by mentioning the results obtained with other time windows and how we decided only to discuss the ones resulting from 6-day time windows, which seem to strike the right balance between reliability and temporal resolution.

- Figure 7, caption: for c, you should mention that f is defined in Figs. 10 and 11.

  **R.**
  Very good observation. We will add the suggested mention.

- page 16, line 11: Is that that in winter the pressure is not showing any daily variations and more a monotonic signal? Any suggestion how the connection between the different boreholes could anyway be qualified?

  **R.**
  Yes, that is the case, and without diurnal variations, our method does not work, a limitation we acknowledge in the method description. We do not have any good suggestions for detecting hydraulic connection in wintertime, as the only distinguishable features on those records are probably related to mechanical processes. An alternative approach could be based on similarities in hydraulic head. However, an equal hydraulic head between two boreholes would be a very weak proof for the existence of a hydraulic connection. The only approach we have considered (but never implemented) is to develop an active sensor that could vary its volume to create small pressure variations. In a confined volume, such pressure variations should be easy to detect by other sensors hydraulically connected. We will add a mention to this idea in the mansucript.

- page 18, line 20: in this chapter?

  **R.**

  We will replace "in this chapter" by "we will present showing"

- page 20, line 6: a small increase

  **R.**

  We will change "an small increase" by "a small increase".

- page 22, lines 12-14: don't really get the point of this isolated paragraph with the surrounding ones.

  **R.**

  It was intended to give context to the following paragraph. However, we acknowledge it seems disconnected; therefore, we will rephrase it and merge it with the following paragraph.

- page 26, line 11: section 2.6

  **R.**

  We will remove the misplaced space within the word "section"

- page 27, line 12: Fig. 5,

  **R.**

  We will replace the misplaced comma by a dot.

- page 32, line 23: resolution of our data?

  **R.**

  We will add the missing "of"

- page 34, line 35: May be you could conclude by a bit of prospective? Which new measurements could allow a better understanding and complement these measurements? Array of seismometers? What more could still be inferred from this existing dataset?

  **R.**

  That is a good suggestion, and it can be combined with the idea of mentioning follow-up papers. We will add a paragraph or two with future work, prospective thoughts and suggestions for new measurements. These paragraphs might also include the mention of the active sensors mentioned above.

---

## Author Response (AR2)

Answers to comments in The Cryosphere Discussion of preprint

**Channelised, distributed, and disconnected: Spatial structure and temporal evolution of the subglacial drainage under a valley glacier in the Yukon**

Camilo Rada G. and Christian Schoof

November 4, 2022

**Contents**

**1 Answers to referee N°1**

**1.1 General comments**

We appreciate your positive comments about the description of the methodology, as well as your constructive feedback regarding the shortcoming of the paper structure and presentation of the results. We restructured the results section as suggested, separating spatial and temporal analysis. We also added a paragraph highlighting the takeaways of each of the four subsections of the discussion. We will answer your numbered list of general suggestions one by one:

1. Regarding the length of the time window, I wonder if using several different time windows with different length would yield more information when comparing their results in term of clustering?

   **R.**
   Due to the reasons explained in our previous response we decided to leave the information of alternative window lengths out of the discussion. However, we added a paragraph mentioning that we decided to leave the information derived from longer and shorter time windows out of the discussion. Noting also that such information increased our confidence in interpreting the six-day time window that we found to strike the right balance between reliability and temporal resolution. See the manuscript with tracked changes on page 12, line 3.

2. I feel that the discussion between correlated and anti-correlated series should be made clearer earlier in the manuscript. The process itself is well illustrated in Figure 1, but I feel that the author are missing an opportunity to clarify their workflow when they introduce the equation for the absolute Euclidian distance where the reason for the use of this specific formulation could be reiterated.

   **R.**
   We agree with the observation. We did introduce figure 7 with an extended explanation right before presenting the absolute Euclidean distance, to clarify the rationale behind this distance choice. See the manuscript with tracked changes on page 12, line 15.

3. At some point in the manuscript, I was not sure if Pressure was designating water pressure or effective pressure, which is a major issue when describing increase or lowering of the pressure. I urge the authors to use either effective pressure or water pressure throughout the manuscript which would help with readability.

   **R.**
   We revised the whole manuscript and use now either "water pressure" or "effective pressure" to avoid any confusion.

4. On the spatial distribution of the disconnected regions, I was wondering if they were appearing consistently in the same region for the different years, and if that is the case, are there any velocity records that they can be compared against?

   **R.**
   As explained in our previous response we made clear that disconnected regions persist through the years with observations. See the manuscript with tracked changes on page 27, line 9. We did also mention that we will discuss their impact on surface speed in a follow-up paper. See the manuscript with tracked changes on page 36, line 35.

**1.2   Specific comments**

- L14-P1: "diffusivity" has a typo.

  **R.**
  Fixed

- L7-P2: The references here all refer to ice-sheets velocity, given the fact that the present study treats of a mountain glacier, references pertaining to this type of glaciers might be better suited.

  **R.**
  We did include several reference to well-studied mountain glaciers. See the manuscript with tracked changes on page 2, line 11.

- L17-P2: "OBP" is defined here but used only once in the text, perhaps it should be omitted and only described in the caption of Figure 1.

  **R.**
  We did remove the OBP acronym and use it only on Figure 1 as suggested,

- L28-P2: The citation of models here is strange, perhaps adding an "e.g" with a shorter list, or a review paper such as de Fleurian et al. (2018); Flowers (2015) would be better suited here.

  **R.**
  We did add e.g. and also Fleurian et al. (2018) in a relevant location in the manuscript. See the manuscript with tracked changes on page 3, line 6.

- L18-P3: "water pressure" should be stated here, or effective pressure (see comment 3 above).

  **R.**
  We did change it to "water pressure"

- Fig 1: Colourblind readers might struggle with the colorscheme of the arrows, perhaps something more contrasted would fit better (gradient of blue to red with black for overburden). In the caption of the figure OBP should be described.

  **R.**
  We did change the colour scheme to the colorblind-friendly choice suggested and described the acronym OBP in the caption.

- L8-P5: It should be "not" not "nor".

  **R.**
  We did change it to "not"

- L27-P5: the recent paper from Doyle et al. (2021) could be cited here too.

  **R.**

  We did cite it as suggested. See the manuscript with tracked changes on page 6, line 6.

- Equation 2: There is an extraneous right parenthesis.

  **R.**

  We did remove the misplaced parenthesis

- L8-P13: It would be nice to have a quick description of the shapes of the pressure record for each cluster here.

  **R.**

  We did add a sentence pointing to their jaggedness and resemblance to a square signal with a peak position just before dropping to base levels. See the manuscript with tracked changes on page 14, line 5.

- L13-P14: The colour coding for correlated and anti-correlated subclusters could be re-iterated here.

  **R.**

  We did add a reiteration of the colour coding of each cluster to that sentence.

- Equation 4: Subscript $i$ is used both for time and the number of valid sample $M_i$ which should be fixed.

  **R.**

  As explained in our previous response this was not an error. However, we did rephrase the paragraph preceding Equation 4 to make this clear. See the manuscript with tracked changes on page 19, line 3.

- Figure 10: I think that clarifying between effective or water pressure is needed in the labels here and in other figures.

  **R.**

  We did change figure 10 and 17 Y axis label to "Normalized water pressure". In this last figure, we did change also "averaged mean pressure" by "averaged mean water pressure" in the caption.

- Figure 10: I expect that the light blue shading is darker when there is snow cover but that should be clarified

  **R.**

  We added a color bar to figure 10, and make it explicit in all caption that figures share the same colour coding for snow cover.

- L4-P19: It should be specified that "the formation of a well developed subglacial drainage system, something that does not occur every year" on this specific site.

  **R.**

  We did change the wording to note that this is a feature particular to South Glacier.

- L5-P20: I have a hard time identifying individual borehole records on Figure 10, perhaps splitting panel a with correlated and anti-correlated borehole in a different panel would help?

  **R.**

  In this figure, we wanted to show the overall pattern of borehole behaviours by type more than providing a good display of individual borehole records. However, this concern is valid, and we increased the contrast between lines to make individual records easier to identify.

- L6-P20: It should be "a" not "an".

  **R.**

  We did change "an small increase" by "a small increase".

- L8-P20: The sentence starting on this line is hard to read and should be rephrased.

  **R.**

  We did rephrase it to: " Such pressure drop would reduce the total normal stress supported by connected areas. Therefore, this unsupported load is transferred to the surrounding unconnected areas where the anti-correlated boreholes are located." See the manuscript with tracked changes on page 20, line 3.

- L15-P20: "through time".

  **R.**

  We did change "trough time" by "through time".

- L33-P20: Perhaps "in the study area" should be added here.

  **R.**

  We did change "that incorporated all the connected sections of the bed." by "that incorporated all the connected sections of the bed under the study area."

- L15-P23: I add to look for the meaning of "straddle" perhaps "intersect" would be better, or am I missing some of the subtleties of the wording?

  **R.**

  We did change "seem to straddle the one on panel f", by "seem to intersect the one on panel f. In a two-dimensional drainage system, such a condition would imply a hydraulic connection between these intersecting clusters. However, the differences in their pressure records suggest that there is no hydraulic connection between them."

- L6-P24: There could be a reference to the section where the probability were introduced here.

  **R.**

  We did add a cross-reference to section 2.5 (Spatial patterns in basal hydraulic connectivity).

- L11-P26: Typo in "section".

  **R.**

  We did remove the misplaced space within the word "section"

- L15-P30: "might be able", "be" is missing.

  **R.**

  We did add the missing "be".

- L24-P30: I am not sure why the discussion on creep that is made below is not stated here.

  **R.**

  We did rephrase to mentioned reductions in volume would be associated with ice creep. See the manuscript with tracked changes on page 32, line 7.

- L11-P32: It should be "boreholes".

  **R.**

  We did replace "holes" by "boreholes".

- L16-P32: The sentence starting on this line is unclear and should be rephrased.

  **R.**

  We did rephase it as: "Such a hydraulic head difference implies that water will flow between two hydraulically connected boreholes. In such a case, we would expect differences in the hydraulic head when significant water storage exists along the flow path. These differences would take the form of oscillations with attenuated amplitude and phase lag". See the manuscript with tracked changes on page 34, line 12.

- L23-P32: "resolution of our data", "of " is missing.

  **R.**
  We did add the missing "of"

- L31-P32: Shouldn't it be "assigns".

  **R.**
  We did change it to "assigns".

- Sup-L33-P2: "reproduce" in place of "reproducing".

  **R.**
  We did change "that does best reproducing" by "that best reproduce".

- Sup-L34-P2: RIG should be defined here.

  **R.**
  We did replace RIG by "Relative Information Gain (RIG)" followed by the reference to the section that describe this concept in depth.

- Sup-L7-P3: EOF should be defined here.

  **R.**
  We did replace EOF by "Empirical Orthogonal Functions (EOF)," followed by the reference.

- Sup-L8-P3: SOMs should be defined here.

  **R.**
  We did replace SOMs by "Self-Organizing Maps (SOMs)," followed by the reference.

- Equation S7 to 9: Shouldn't it be $a_i$ in place of $a_1$

  **R.**
  We did replace all $a_1$ and $b_1$ by $a_i$ and $b_i$ respectively.

- Sup-L6-P14: "mechanical".

  **R.**
  We did replace the typo "machanical" by "mechanical"

**2  Answers to referee N°2**

**2.1  General comments**

We appreciate your positive comments about our manuscript and the uniqueness of the presented dataset. We were intentionally cautious of not explicitly saying that the dataset is unique and exceptional. However, your comments encourage us to put some of that caution aside. Therefore, we made some changes to the abstract to better communicate the uniqueness of the presented dataset to the reader. See the manuscript with tracked changes on page 1 line 5. We did also emphasize the previous publications discussing this dataset and mention that a follow-up papers are coming to explores the effects of the subglacial drainage evolution on surface speed variations. See the manuscript with tracked changes on page 36, line 35.

**2.2  Minor remarks**

- page 3, line 18: about the winter measurements, if most boreholes display pressure near overburden pressure for several months, one should expect high or even increasing surface velocity? Do you have observation of surface velocity all around the year that show that? May be this link with the surface velocity measurements (as mentioned in 2.1) even if I understand you don't have surface velocity in 2015?

  **R.**
  As explained in our previous answers. We do have continuous surface velocity records for the whole

studied period, including 2015. However, due to the extent of the manuscript we have intentionally left out the analysis of velocity data in the context of the described temporal evolution of the subglacial drainage system. That will be the main focus of a follow-up paper. We made the rationale explicit in the text and mention that the above paper is in preparation.

- page 3, line 19: which size are expected to be this "water pockets"? The wording make me think to a feature that as similar vertical and horizontal dimensions, whereas I expect more a flat feature? May be "water patches" would be more appropriate here?

  **R.**
  Following our previous answer, we now explicitly mention that we do not have constraints on the exact shape and size of these water pockets and explain exactly what we mean when using this terminology. See the manuscript with tracked changes on page 3, line 26.

- page 3, line 20: sometime you are using upper case after a colon, sometime not. Here I would said that a starting a new sentence would work better as the second point is at the beginning of a new sentence (even a new paragraph).

  **R.**
  We did check the whole manuscript to make sure there is no capitalization after colons, except for proper nouns and acronyms. In this case, we did start a new sentence and removed the paragraph break before describing the second process.

- Figure 1: cases b, c, d are not steady, in comparison to a that can be steady. May be it should be mentioned in the caption. Also, how long do we expect these unsteady situations to last? Give order of magnitude.

  **R.**
  Following our explanation in our previous answer, we did add that a sentence to the caption saying: "Note that (a) is a stable configuration and (d) is unstable, while the stability of (b) and (c) depends on the conditions".

- page 4, line 3: missing a verb in the last part of the sentence?

  **R.**
  We did review the wording.

- page 4, line 12: it is obvious, but may be you should mention here that you are measuring water pressure in these boreholes?

  **R.**
  We did add a mention to the fact that what we are actually measuring in the boreholes is water pressure.

- page 11, line 7: how sensitive are your results to this choice of a 6-day time window?

  **R.**
  Following on the responses to referee N°1, we will elaborate by mentioning the results obtained with other time windows and how we decided only to discuss the ones resulting from 6-day time windows, which seem to strike the right balance between reliability and temporal resolution. See the manuscript with tracked changes on page 12, line 3.

- Figure 7, caption: for c, you should mention that f is defined in Figs. 10 and 11.

  **R.**
  We did add the suggested mention.

- page 16, line 11: Is that that in winter the pressure is not showing any daily variations and more a monotonic signal? Any suggestion how the connection between the different boreholes could anyway be qualified?

  **R.**
  Following our previous answer, we did elaborate on how it might be possible to detect connection on such cases. See the manuscript with tracked changes on page 37, line 6.

- page 18, line 20: in this chapter?

  **R.**

  We did replace "in this chapter" by "we will present showing"

- page 20, line 6: a small increase

  **R.**

  We did change "an small increase" by "a small increase".

- page 22, lines 12-14: don't really get the point of this isolated paragraph with the surrounding ones.

  **R.**

  We did rephrase it and merge it with the following paragraph. See the manuscript with tracked changes on page 23, line 16.

- page 26, line 11: section 2.6

  **R.**

  We did remove the misplaced space within the word "section"

- page 27, line 12: Fig. 5,

  **R.**

  We did replace the misplaced comma by a dot.

- page 32, line 23: resolution of our data?

  **R.**

  We did add the missing "of"

- page 34, line 35: May be you could conclude by a bit of prospective? Which new measurements could allow a better understanding and complement these measurements? Array of seismometers? What more could still be inferred from this existing dataset?

  **R.**

  We did add a paragraph with future work, prospective thoughts and suggestions for new measurements. See the manuscript with tracked changes on page 37, line 3.